# AD-NODE: Adaptive Dynamics Learning with Neural ODEs for Mobile Robot Control

## Abstract

Mobile robots, such as ground vehicles and quadrotors, are becoming increasingly important in various fields, from logistics to agriculture, where they automate processes in environments that are difficult to access for humans. However, to perform effectively in uncertain environments using model-based controllers, these systems require dynamics models capable of responding to environmental variations, especially when direct access to environmental information is limited. To enable such adaptivity and facilitate integration with model predictive control, we propose an adaptive dynamics model which bypasses the need for direct environmental knowledge by inferring operational environments from state-action history. The dynamics model is based on neural ordinary equations, and a two-phase training procedure is used to learn latent environment representations. We demonstrate the effectiveness of our approach through goal-reaching and path-tracking tasks on three robotic platforms of increasing complexity: a 2D differential wheeled robot with changing wheel contact conditions, a 3D quadrotor in variational wind fields, and the Sphero BOLT robot under two contact conditions for real-world deployment. Empirical results corroborate that our method can handle temporally and spatially varying environmental changes in both simulation and real-world systems.

## 1 Introduction

Mobile robots such as ground vehicles and quadrotors are increasingly used across applications, from navigating warehouse floors in logistics to large-scale crop monitoring in agriculture (Li et al., 2024; Duggal et al., 2016). These systems provide access to environments difficult to access for humans, enabling greater operational scale and improved efficiency. Mobile robots may encounter various types of environments within a single operation and generally lack prior knowledge of the conditions they will face, making real-time adaptability essential during deployment. To operate robustly in such uncertain environments, mobile robots require adaptive control strategies that can respond to environmental variations such as terrain types, wind conditions, or payload changes. However, achieving such adaptability remains challenging for model-based controllers as they rely on accurate dynamics models for control action planning over long horizons (Seo et al., 2020; Nagabandi et al., 2018a). Furthermore, many environmental variations cannot be fully detected using onboard sensors, making it important for the system to infer hidden environmental factors from limited data and adapt its dynamics accordingly.

Previous efforts in in-context reinforcement learning (RL) have led to major advances in adapting to different environments based on past trajectories (Liang et al., 2023; Zhang et al., 2025; Belkhale et al., 2021). A line of research in adaptive model-free RL proposes specially designed adaptive modules, known as Rapid Motor Adaptation (RMA), to encode environmental information in RL policy (Kumar et al., 2021; Zhang et al., 2023; Qi et al., 2023). However, model-free RL methods often struggle to explicitly incorporate desired trajectories or hard constraints, such as collision avoidance, into the policy, which in turn requires large amounts of exploratory data. Several model-based RL approaches have been proposed (Seo et al., 2020; Lee et al., 2020; Evans et al., 2022); however, they often model dynamics over a predefined discrete time domain, which overlooks the continuously-evolving dynamics of rigid-body robotic systems (Greydanus et al., 2019). Since the dynamics of these systems are typically governed by ordinary differential equations (ODEs), neural ordinary differential equations (NODE) (Chen et al., 2018), which learn first-order derivatives and

Figure 1: (a) Our adaptive dynamics model outperforms CaDM (Lee et al., 2020) when combined with MPC in goal-reaching and path-tracking tasks across (top row) differential wheeled robot and (bottom row) quadrotor navigation platforms. Our method works well in unknown environments (such as different layouts of surface textures and wind fields) and accurately reaches the targets, while CaDM struggles with oscillations around the targets. (b) Physical setup of a Sphero BOLT robot navigating through different textures and reaching the goal.

compute system states using numerical integrators, are well-suited for modeling continuous-time dynamics. The approach of modeling derivatives has also shown success in time-series prediction tasks across various domains (Lipman et al., 2022; Zhang et al., 2024; Cranmer et al., 2020). In robotics, learning dynamics with NODE has demonstrated robustness to noisy and irregular data in standard RL tasks (Yildiz et al., 2021). However, the effectiveness of continuous-time models for capturing adaptive dynamics under drastic environmental changes remains an open question. In this work, we propose AD-NODE, an adaptive dynamics model for mobile robots that combines NODE with an adaptive module in the style of RMA to infer environmental conditions from past state-action history. We use a two-phase training framework: in Phase 1, the model focuses on learning the mapping between states, with complete environmental information (referred to as "privileged information") included in the training data. In Phase 2, the model learns to reconstruct the environment from historical data during execution. The proposed adaptive dynamics model is used with model predictive control (MPC) (Morari & Lee, 1999; Garg et al., 2013; Chua et al., 2018) to determine the optimal actions for accomplishing navigation tasks on two simulated mobile robotic platforms: a 2D differential wheeled robot navigating surfaces with different textures, and a 3D quadrotor flying through different wind fields. Given the limited availability of models that are both adaptive and continuous, and with the goal of enabling adaptability in mobile robots across varying environments, we select a classic context-aware dynamics model (CaDM) (Lee et al., 2020) as our primary comparison baseline. Figure 1 demonstrates that our proposed model has superior performance in both goal-reaching and path-tracking tasks across both simulated platforms. Furthermore, the model we design can be deployed in a real-world environment where a Sphero BOLT robot navigates across two distinct textures.

## 1.1 CONTRIBUTION

We propose learning a continuous-time adaptive dynamics model with NODE (AD-NODE) for mobile robotic systems that can adapt to the environment during operation. Specifically:

- We propose a novel framework that incorporates adaptability into continuously-evolving dynamics for long-horizon rollouts in MPC.

- We empirically show that our framework achieves higher accuracy compared to the baselines in both goal-reaching and path-tracking tasks for differential wheeled robot and quadrotor navigation platforms.

- We validate the feasibility of AD-NODE beyond simulation by deploying it on a Sphero BOLT robot across surfaces with different friction, demonstrating adaptability and repeatability under hardware uncertainty.

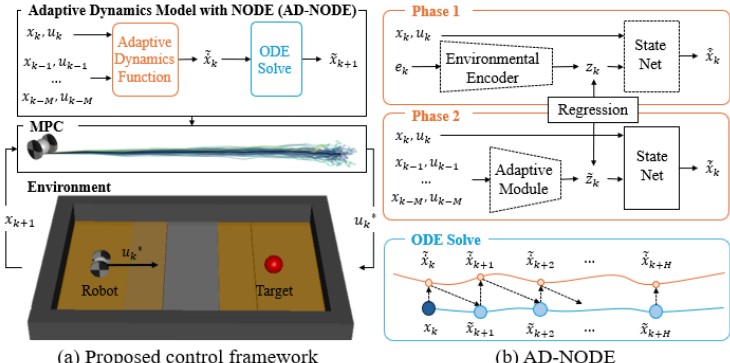

Figure 2: (a) Proposed control framework for mobile robots, where MPC is adopted to determine optimal control actions by predicting future trajectories with our proposed dynamics model (AD-NODE). (b) Structure of AD-NODE: the state net models the derivatives of states evolution, the environmental encoder processes privileged information, and the adaptive module reconstructs a latent environmental vector from historical state-action data by regressing to the corresponding latent vector from Phase 1. State prediction is obtained through numerical integration of the dynamics function. Models with trainable weights are indicated with dashed lines.

## 2 RELATED WORK

Adaptive dynamics modeling for robotic systems has been explored through a variety of approaches aimed at handling changing and partially observed environments. A common strategy uses historical state-action sequences to infer latent vectors representing the environment, which then inform the dynamics model. Different model structures have been used for this purpose, including transformers (Xiao et al., 2024) for handling long sequences and capturing temporal dependencies, graph neural networks for object-centric embeddings (Li et al., 2019), and probabilistic models to account for uncertainty in partially observed information (Guttikonda et al., 2024; Belkhale et al., 2021). Beyond model structures, various training pipelines have been explored, such as using both a forward and a backward path (Lee et al., 2020), meta-learning approaches that update themselves across a distribution of environments (Nagabandi et al., 2018a; Levy et al., 2025), and learning a multi-modal model that selects the correct mode on the fly (McKinnon & Schoellig, 2017). Some research also integrates multi-modal inputs, embedding heterogeneous sensor signals into latent spaces to enhance dynamics predictions (Vertens et al., 2023). Inspired by adaptive control theory, other approaches capture environmental variations by modeling residual errors from a reference model using low-pass filters (Huang et al., 2023; Hanover et al., 2021). Finally, continual online learning methods have been proposed to maintain adaptability by updating the model with new information during operation (Jiahao et al., 2023; Nagabandi et al., 2018b). Our work builds on these foundations by using environmental labels to guide historical state–action processing as well as structuring the model with a NODE backbone, yielding a practical dynamics model that works across different robotic platforms and environmental conditions.

## 3 PROBLEM STATEMENT

We address the problem of goal reaching or path tracking of mobile robots as a discrete-time MPC, which optimizes a sequence of future control actions over a finite time horizon. The dynamical system we are considering is governed by

$$\dot{x}(t) = f(x(t), u(t), e(t)), \quad x_{k+1} = x_k + \int_{t=k}^{t=k+1} f(x(t), u(t), e(t)) \, dt \tag{1}$$

where $x_k \in \mathbb{R}^n$ denotes the state of the system at time step $k$, $u(t) \in \mathbb{R}^m$ is the control input, $e(t) \in \mathbb{R}^l$ represents environmental factors that will influence the dynamics, and $f$ represents the continuous-time dynamics of the system that models $\dot{x}(t)$. $x_{k+1}$ can be obtained using a numerical

integrator such as *Euler* or *Runge–Kutta methods*. After discretization, $u(t)$ is represented as $u_k$, and $e(t)$ as $e_k$.

Over a prediction horizon $H$, the task can be formulated as an optimization problem that can be solved using MPC (Morari & Lee, 1999; Garg et al., 2013; Chua et al., 2018) at each time step $k$:

$$
\min_{u_{k:k+H-1}} \quad \sum_{i=k}^{k+H-1} \ell(x_i, u_i) + \ell_f(x_{k+H})
$$
$$
\text{s.t.} \quad x_{i+1} = \text{ODESolve}(x_i, u_i, f, t_i, t_{i+1}), \quad i = k, \ldots, k+H-1,
$$
$$
u_i \in \mathcal{U}, \ x_0 \in \mathcal{X}_0
$$
(2)

where $\ell$ is the stage cost, $\ell_f$ is the terminal cost. $\mathcal{U}$ denotes the input constraint sets, and $\mathcal{X}_0$ is an initial constraint set designed to ensure a robot starts from the designated state. After solving Equation 2, only the first control input $u_k^\star$ of the optimal sequence is applied to the system. In the next time step, the horizon is shifted forward, and the optimization is repeated with updated state information.

With a fixed dynamics model, MPC can handle a certain degree of uncertainty or disturbances via solving Equation 2 on the fly during execution. However, it fails to handle systems that deviate too much from the reference dynamics model, such as through unexpected environmental changes or severe disturbances. To enable fast and effective adaptability, we propose to adapt the dynamics function $f$ by updating the environment ($e_k$) between control steps. However, it is often hard to find the complete environment information because the real-world system is often partially observed. Therefore, the state-action history is used to recover the current environment. Inspired by RMA (Kumar et al., 2021), we first encode environmental factors $e_k$ into an environment latent vector $z_k$ and learn an adaptive module to encode state-action history into latent vector $\tilde{z}_k$. Then, the environment can be recovered by training encoders to align $\tilde{z}_k$ and its corresponding $z_k$ together. The process can be expressed as

$$
z_k = g(e_k), \quad \tilde{z}_k = h(\{(x_i, u_i)\}_{i=k-M}^{k-1}), \quad L = L_2(z_k, \tilde{z}_k),
$$
(3)

where $\{(x_i, u_i)\}_{i=k-M}^{k-1}$ denotes state-action history over a horizon of length $M$. $g$ denotes the environmental encoder that encodes $e_k$ to $z_k$, and $h$ denotes the adaptive module that reconstructs current environment by encoding state-action history to $\tilde{z}_k$ and regressing to $z_k$ by MSE loss. We recover the environment in latent space because it is easier to align two different domains (the domain of $\{(x_i, u_i)\}_{i=k-M}^{k-1}$ and its corresponding $e_k$) in another lower-dimensional space. See Figure 2 for the complete framework.

## 4 ADAPTIVE DYNAMICS MODEL WITH NODE

This section proposes an adaptive dynamics model with NODE (AD-NODE) for mobile robots, which integrates environment-aware dynamics into MPC to obtain optimal actions for navigation.

### 4.1 TWO-PHASE FRAMEWORK FOR LEARNING ADAPTIVE DYNAMICS

In this section, we discuss how to learn the dynamics mapping from the current state $x_k$ and current action $u_k$ to the next state $x_{k+1}$ conditioned on $M$ historical data $\{(x_i, u_i)\}_{i=k-M}^{k-1}$. The objective of learning the adaptive dynamics is to capture task-invariant environments based solely on historical data (Equation 3), so that the dynamics function can be adjusted according to the inferred environment. While most model-based RL learns the mapping in an end-to-end manner, we decompose dynamics learning into two phases as shown in Figure 2.

In Phase 1, we learn state evolution using privileged information $e_k$, which is available in simulation but may not be measurable during deployment. Conditioning on $e_k$, which carries direct and complete environmental information, facilitates learning the state evolution in response to control actions. The state evolution is implemented in the state net with NODE to learn the first derivative

$\dot{x}(t)$, and numerical integrators are used to compute the next state $x_{k+1}$. This process models trajectories as integration of the vector fields, inherently producing smooth and physically consistent outputs. A.1 and A.4.3 present both theoretical and empirical evidence for the advantages of modeling dynamics with a NODE over a Multi-Layer Perceptron (MLP). However, instead of inputting $x_k$, $u_k$, and $e_k$ directly into the state net, we apply an environmental encoder to first turn $e_k$ into a latent vector $z_k$, which is then passed to the state net.

After completing end-to-end training in Phase 1, Phase 2 addresses the original mapping problem by learning to infer $z_k$ from historical data. Since Phase 1 is dedicated to modeling the temporal evolution of system states, we fix the weights of the learned state net in Phase 2. We then regress the historical data on their corresponding $z_k$ and obtain $\tilde{z_k}$ by following the process mentioned in Equation 3 and Figure 2. Since historical data carries distinct physical meanings and have significantly different dimensionality compared to $e_k$, it is effective to align two domains together in latent space. A similar approach has been proven successful in style transfer, where a domain-invariant representation is learned in latent space to facilitate knowledge transfer or generate consistent outputs across different styles (Gatys et al., 2016).

## 4.2 Sampling-Based Control with Online Dynamics Learning

NODE results in strong extrapolation capabilities and temporal continuity, making it particularly well-suited for integration with sampling-based MPC: in particular, we use the model predictive path integral (MPPI) framework (Williams et al., 2017). Within the MPPI framework, a large set of control sequences $\{u_{k:k+H-1}^{(i)}\}_{i=1}^{N}$ are sampled and propagated forward using Equation 1 to generate corresponding trajectories. The cost of each trajectory is evaluated using a task-specific cost function $J^{(i)}$ consisting of stage cost $l$ and terminal cost $l_f$, and the optimal control is computed as a weighted average $u_k^* = \sum_{i=1}^{N} w^{(i)} u_k^{(i)}, \quad w^{(i)} = \frac{\exp\left(-\frac{1}{\lambda} J^{(i)}\right)}{\sum_{j=1}^{N} \exp\left(-\frac{1}{\lambda} J^{(j)}\right)}$, where $\lambda$ is a temperature parameter controlling exploration. A.2 shows the convergence analysis of using Phase 2 model as dynamics function within MPC framework.

To improve performance across environments, we incorporate online fine-tuning of the learned dynamics model. As new observations $\{(x_k, u_k, x_{k+1})\}$ become available during execution, we update the parameters in the dynamics model on the fly to reduce prediction errors. To avoid catastrophic forgetting as well as to balance exploration and exploitation, we use experience replay buffers to record all the observations for online learning and enable a robot to select between a random action and $u_k^*$. This continual learning process allows the model to refine its predictions and adapt to distributional shifts, especially for unseen historical data.

## 4.3 Additional Training Details

To improve long-horizon prediction accuracy, we apply curriculum learning to train NODE with gradually increasing prediction horizons, from 1-step to H-step alignment. This mitigates the gradient explosion and vanishing issues that commonly arise when training on long sequences or fine-tuning an existing model. In addition, for the adaptive module design, using a 1D convolutional neural network (CNN) to handle the high dimensionality of historical data shows benefits in quadrotor experiments by improving latent vector reconstruction through the extraction of local temporal patterns. The sliding filters capture environmental changes regardless of their position in the sequence, which is useful in robotic motion, where accelerations or directional shifts can occur at arbitrary points within a time sequence. Since each element in the state and action vectors represents a distinct physical quantity, treating them as separate channels further enhances feature extraction.

# 5 Simulation Experiments

## 5.1 Baseline Methods

**MLP-based dynamics** We consider two MLP-based dynamics: Phase 1 uses privileged information and trains the model autoregressively with an L2 loss over a fixed horizon to assess the benefits of NODE, and Phase 2 builds on the Phase 1 model by incorporating historical data for partially observed settings.

**Context-aware dynamics model (CaDM)** Lee et al. (2020) trains the adaptive dynamics in an end-to-end manner by using forward and backward loss to extract environmental factors. The comparison evaluates the overall adaptability of the learned dynamics model under environmental changes.

**Meta-learning based dynamics model** We adopt the concept of the meta-learning–based approach in Belkhale et al. (2021) to design a context encoder using variational inference. The comparison evaluates the overall adaptability of the learned dynamics model under environmental changes.

**Fixed NODE-based dynamics** A NODE-based model that does not update environmental information during operation and relies solely on the initial environmental factors for inference.

**AD-NODE** AD-NODE is developed in two phases: Phase 1 is trained with privileged information and serves as an upper bound for performance; it uses NODE and is solved with the *Forward Euler method*. Phase 2 builds on the Phase 1 model by incorporating historical state-action information through an encoder that reconstructs $z_k$ for partially observed settings.

**RMA** (Kumar et al., 2021) RMA is a representative model-free method in this work. To enable trajectory tracking capability, we encode desired future positions into a embedding vector and incorporating it to the Proximal Policy Optimization (PPO) policy (Schulman et al., 2017).

**DATT** (Huang et al., 2023) DATT is a model-free method that uses L1 adaptation to derive the environmental factors. Training is also based on PPO algorithm (Schulman et al., 2017).

## 5.2 DATA COLLECTION

We collect a dataset using the simulator by performing a grid search over the span of the state and action spaces in different environmental factors. Note that fully covering these spaces is challenging, especially for the high-dimensional quadrotor system. Therefore, we adopt a coarse sampling strategy: for the differential wheeled robot, we grid-sample the initial states and choose a random action but fix it along a trajectory; for the quadrotor, we randomly sample initial states and apply random actions. Each trajectory consists of 50 state-action pairs. If the dynamics model trained on the dataset fails to accurately capture the true system behavior, the online dynamics learning approach discussed in Section 4.2 becomes essential as it helps the robot explore and cover the critical portions of the state-action space required to accomplish the task.

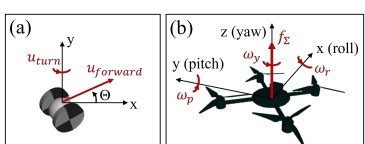

Figure 3: Environment setup for (a) 2D differential wheeled robot and (b) 3D quadrotor.

## 5.3 2D DIFFERENTIAL WHEELED ROBOT NAVIGATION

**Setup** Instead of relying on an equation-based simulator, we implement the environment using MuJoCo physics engine (Todorov et al., 2012) to provide more realistic simulations for a contact-rich environment. The environment features a mobile robot with two cylindrical wheels operating on a 2D surface. The robot's state is defined as $[x, y, \theta, \dot{x}, \dot{y}, \omega]^T$, where $x$ and $y$ represent the position, $\dot{x}$ and $\dot{y}$ represent the corresponding velocity, and $\theta$ and $\omega$ denote the heading and angular velocity (Figure 3). The robot is driven by a differential drive, which allows the wheels to rotate at different speeds and directions, resulting

Table 1: Performance of the differential wheeled robot under *(i)* spatially continuous friction. Success rate (%) for goal-reaching and position RMSE (m) for path-tracking.

|  | Goal-reaching | Path-tracking |
|---|---|---|
| MLP(Phase1) | 2 | > 0.1 |
| MLP(Phase2) | 2 | > 0.1 |
| CaDM | 16 | > 0.1 |
| Meta-learning based | 10 | 0.053±0.028 |
| Fixed NODE | 74 | 0.045±0.015 |
| AD-NODE(Phase1) | 76 | 0.026±0.013 |
| AD-NODE(Phase2) | **98** | **0.021±0.012** |

in two control actions: forward velocity command and steering angle command, represented as $[u_{\text{forward}}, u_{\text{turn}}]^T$. Our proposed MPC controller will determine the $u_{\text{forward}}$ and $u_{\text{turn}}$ and a low-level PID controller with fixed parameters is used to transform the high-level command to low-level motor torque commands for each wheel.

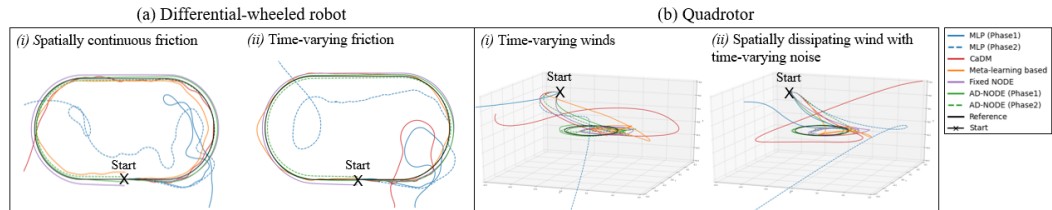

Figure 4: Path tracking trajectories of AD-NODE and model-based baselines in (a) differential-wheeled robot and (b) quadrotor simulators. The time-varying friction is updated every 5 control steps, and the time-varying wind field is updated every 10 control steps.

Table 2: Performance of the differential wheeled robot under *(ii)* time-varying friction updated every 5, 10, or 20 control steps. Success rate (%) for goal-reaching and position RMSE (m) for path-tracking.

| | Goal-reaching | | | Path-tracking | | |
|---|---|---|---|---|---|---|
| | 5 steps | 10 steps | 20 steps | 5 steps | 10 steps | 20 steps |
| MLP(Phase1) | 2 | 2 | 0 | > 0.1 | > 0.1 | > 0.1 |
| MLP(Phase2) | 0 | 0 | 2 | > 0.1 | > 0.1 | > 0.1 |
| CaDM | 12 | 10 | 14 | > 0.1 | > 0.1 | > 0.1 |
| Meta-learning based | 16 | 14 | 18 | 0.030±0.021 | 0.040±0.022 | 0.054±0.033 |
| Fixed NODE | 66 | 62 | 60 | 0.038±0.012 | 0.035±0.015 | 0.044±0.014 |
| AD-NODE(Phase1) | 74 | 76 | 74 | **0.024±0.015** | **0.024±0.015** | **0.025±0.016** |
| AD-NODE(Phase2) | **92** | **94** | **94** | 0.024±0.014 | 0.028±0.016 | 0.031±0.018 |

Environmental variations are surface textures, characterized by sliding, turning, and rolling friction, denoted as $\mu_{\text{sliding}}, \mu_{\text{turning}}, \mu_{\text{rolling}}$. We assume isotropic friction for simplicity. During data collection, we pick two surface textures, one is easier to maneuver on, and the other is slippery and requires more energy to move forward. See A.3.1 for simulation details. We collect trajectories on each surface for training. During testing, the robot is expected to detect surface changes and generalize to cases where the friction varies between the training values, as well as to cases where its wheels contact both surfaces simultaneously.

**Target tasks** Figure 1 visualizes the two tasks under piecewise constant friction. In the goal-reaching task, the differential wheeled robot starts from the left or right boundaries and uses a controller to reach the target at the center, with varying initial headings. This set-up evaluates performance across the entire domain. Success rate is defined as the percentage of episodes in which the agent reaches the target within a 10 mm threshold and within 150 control steps, measured over 50 runs. In the path-tracking task, the robot follows a predefined stadium-shaped path, with performance measured by the RMSE between the robot's position and the target path over a complete lap. For both tasks, the test environments are evaluated in two environments: *(i)* spatially continuous friction that changes with radial distance from the environment center; and *(ii)* time-varying friction, updated every 5, 10, or 20 control steps. See A.6.2 for the cost functions used in MPC.

**MPC performance** In Table 1 and 2, we observe that our proposed model consistently outperforms the baselines, achieving higher goal-reaching success rates and lower path-tracking errors under both environments. Intuitively, the Phase 1 model is expected to perform better due to access to privileged information; however, Phase 2 achieves higher success rates in the goal-reaching tasks. This may be because Phase 1 training does not include scenarios where the two wheels are on different surfaces, whereas the Phase 2 model, trained on historical data, may generalize better to such cases. For path-tracking tasks, both Phase 1 and Phase 2 variants of our model show comparable performance across environments. Figure 4 shows the robot trajectories in the path-tracking tasks. The MLP-based dynamics models fail to complete the lap, CaDM completes most of it, and both the meta-learning-based and fixed-dynamics models successfully finish the full lap. Our proposed model achieves the best reference-tracking performance, consistent with the results reported in the tables.

Table 3: Performance of the quadrotor is evaluated under *(i)* time-varying winds fluctuating sinusoidally around nominal, updated every 10 control steps, and *(ii)* spatially dissipating wind with random time-varying noise modeled as Brownian motion. Shown as RMSE (m).

| | *i* | | *ii* | |
|---|---|---|---|---|
| | Goal-reaching | Path-tracking | Goal-reaching | Path-tracking |
| MLP(Phase 1) | $> 0.2$ | $> 0.2$ | $> 0.2$ | $> 0.2$ |
| MLP(Phase 2) | $> 0.2$ | $> 0.2$ | $> 0.2$ | $> 0.2$ |
| CaDM | $> 0.2$ | $> 0.2$ | $> 0.2$ | $> 0.2$ |
| Meta-learning based | 0.092±0.023 | 0.143±0.074 | 0.099±0.021 | 0.119±0.066 |
| Fixed NODE | 0.066±0.011 | 0.055±0.019 | 0.065±0.008 | 0.049±0.013 |
| RMA | 0.107±0.012 | 0.099±0.014 | 0.065±0.015 | 0.062±0.012 |
| DATT | 0.083±0.016 | 0.070±0.013 | 0.051±0.008 | 0.048±0.009 |
| AD-NODE(Phase 1) | **0.010±0.004** | **0.036±0.020** | **0.012±0.007** | 0.034±0.012 |
| AD-NODE(Phase 2) | 0.031±0.016 | 0.049±0.026 | 0.022±0.014 | **0.030±0.018** |

## 5.4 3D QUADROTOR TRAJECTORY PLANNING

**Setup** We use the quadrotor dynamics as the governing equation in our simulator (Huang et al., 2023). Since we are only concerned with the position, velocity and orientation of a quadrotor, the state that can capture the simplified dynamics model is defined as $[p, v, q]^T$, where $p$, $v$ and $q$ are the 3D position, velocity, and quaternion. As shown in Figure 3, control action is defined as $[f_\Sigma, \omega]^T$, where $f_\Sigma$ denotes the desired thrust and $\omega$ denotes the desired angular velocity in roll, pitch and yaw direction $[\omega_r, \omega_p, \omega_y]$. The MPC controller will determine the high-level actions, thrust and angular velocity, and we implement low-level controller including a PI controller that transforms the high-level commands to thrust and torque followed by an inverse mapping of an actuation matrix for obtaining the four motor speeds. To model a real-world quadrotor, we transform the continuous dynamics model into a discrete one with the sampling time set as 0.02 seconds.

Environmental variations are different wind fields acting on the quadrotor system. The wind field is modeled as a disturbance force along the x-direction. During data collection, we sample the disturbance force in the range of $[-1, 1]$ N and collect trajectories in each wind field. During testing, the quadrotor is subjected to wind fields beyond the $[-1, 1]$ N range, representing out-of-distribution environmental conditions.

**Target tasks** Figure 1 visualizes the two tasks under piecewise constant wind field. In the goal reaching and hovering task, the goal is at the origin, and the quadrotor can start from each vertex of a cube where the cube's edge length is 0.4 meters long. The objective is to control the quadrotor to reach and hover at the goal. We allow each controller 5 seconds to execute actions, then calculate the average position error (RMSE) between the quadrotor position and goal over the final second. In the path tracking task, the objective is to track a circle starting from the same positions as in the goal reaching tasks, using the same RMSE metric. Test environments include two wind conditions: *(i)* time-varying wind fluctuating sinusoidally around a nominal force, updated every 10 control steps; and *(ii)* spatially dissipating wind with random time-varying noise modeled as Brownian motion (Huang et al., 2023), which represents the wind field of a fan blowing on a quadrotor. See A.6.2 for the cost function used in MPC.

**MPC performance** Table 3 shows the results of the two tasks in both environments after integrating the learned dynamics model with MPC. We also apply online dynamics learning, as described in Section 4.2, to fine-tune the dynamics model (see A.6.3 for details). Similar to Section 5.3, our approach successfully completes each navigation task in environments that vary across space and time, achieving lower tracking errors than the model-based baselines. While there is a slight performance drop from Phase 1 (with privileged information) to Phase 2, the adaptive module in Phase 2 still reconstructs environmental factors and outperforms both the fixed NODE-based dynamics model and other model-based baselines. The quadrotor trajectories in the path-tracking task (Figure 4 ) show that the meta-learning-based and fixed-dynamics models perform best among the baselines, successfully guiding the quadrotor near the target trajectory. However, these top baselines still cannot track the reference accurately, whereas AD-NODE achieves the best overall tracking performance.

Table 3 also compares model-free (RMA and DATT) and model-based approaches. These two categories use fundamentally different training pipelines: PPO, a model-free and on-policy method,

trains the policy directly from data sampled by the current policy, while our model-based approach first collects offline data using random policies to train a dynamics model, then fine-tunes the model using online data generated by the MPPI controller. Because of these differences, it is difficult to compare the methods using exactly the same training dataset. Nevertheless, we report the results in Table 3 and show that our approach achieves comparable performance. The qualitative results of the model-free baselines can be found in A.8.

Model-free methods may underperform partly due to how desired trajectories are incorporated during PPO training. MPPI naturally embeds the desired trajectory into its cost function, so the dynamics model does not need to learn trajectory-specific behavior. In contrast, PPO must sample trajectories during training, and these samples may not fully cover the evaluation trajectories. MPPI also allows soft or hard constraints (e.g., collision avoidance) to be directly added to the optimization, improving safety. On the other hand, model-free methods offer significantly faster inference: PPO runs at roughly $0.001$ s per step—an order of magnitude faster than our $0.01$ s—making them attractive for applications requiring very high control rates or limited onboard compute.

## 6 REAL-WORLD EXPERIMENTS

### 6.1 SETUP

To demonstrate the effectiveness of deployment on real-world robots, we evaluate the proposed framework on a mobile robot platform (shown in Figure 5 (a)) where a Sphero BOLT robot is operated on a pool table and a Luxonis Oak-D Pro camera is mounted above the table to detect robot state. Although the robot has a spherical outer shape, its internal actuation resembles a differential-drive mechanism, so it cannot move sideways without first turning its heading. Therefore, this robot is a suitable platform for testing planning algorithms for navigation tasks.

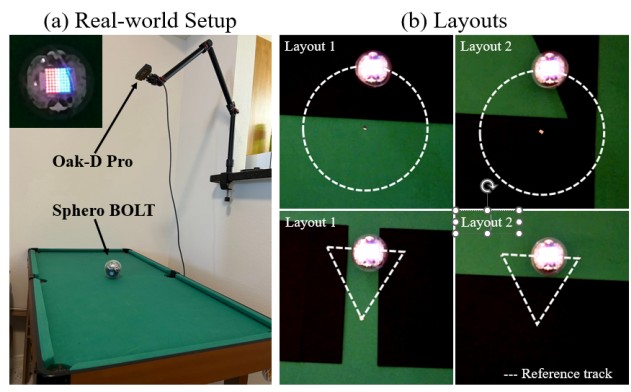

Figure 5: (a) Environment setup for the real-world platform. (b) Friction layouts for path tracking.

The state of the robot is defined as $[x, y, dx, dy, \dot{x}, \dot{y}]^T$, where $x$ and $y$ represent the 2D position, $dx$ and $dy$ denote the cosine and sine of the robot heading, $\dot{x}$ and $\dot{y}$ represent the corresponding velocity. We use the frame capture loop of the camera, running at 60 fps with a frame step size of $\Delta t = 0.0167$ seconds, and obtain a control frequency of 15 Hz and control step size of $dt = 0.0667$ seconds, by applying action every 4 frame capture loop. To estimate the robot state, two colored patches (Figure 5 (a), top left) are displayed on the LED array, and a blob detection algorithm identifies their centers $p_1 = (u_1, v_1)$ and $p_2 = (u_2, v_2)$. Due to the symmetric design of two patches, the center of the robot $(x, y)$ can be calculated by the average of $p_1$ and $p_2$. To avoid ambiguity in the angle of the heading, we represent the heading with two entries $(dx, dy)$, which are calculated by $(u_1 - x, v_1 - y)$. To estimate the robot's velocity $(\dot{x}, \dot{y})$, we record the robot center $(x_{prev}, y_{prev})$ two frames before each control step. The velocity is then computed as $((x - x_{prev})/(2\Delta t), (y - y_{prev})/(2\Delta t))$.

The action of the robot is defined as $[u_{\text{forward}}, u_{\text{turn}}]^T$, which represent the forward velocity command and the steering angle command respectively. The proposed framework will determine the optimal action at each control step and the internal low-level controller in the Sphero BOLT robot will track the commanded action. We note that the actual run time of obtaining an optimized action from the framework is less than $0.01$ seconds on a single Nvidia RTX 3090 GPU. The reason we slow down the control rate is to accommodate the blob detection algorithm and the Bluetooth communication, which take most of the time. To evaluate the capability of adapting to different environments, we

Table 4: Performance of real-world experiments on navigation tasks with different friction layouts. Shown as RMSE (cm).

| | Goal-reaching | | Circle-tracking | | Triangle-tracking | |
|---|---|---|---|---|---|---|
| | Layout 1 | Layout 2 | Layout 1 | Layout 2 | Layout 1 | Layout 2 |
| Fixed NODE-based | 2.127 ± 1.233 | 2.725 ± 1.176 | 4.567 ± 2.106 | 2.858 ± 0.744 | 5.404 ± 0.193 | 5.124 ± 0.318 |
| AD-NODE (Phase 2) | **0.766 ± 0.303** | **0.372 ± 0.082** | **2.077 ± 0.340** | **2.417 ± 0.124** | **3.315 ± 0.027** | **2.681 ± 0.113** |

create different friction layouts (shown in Figure 5 (b)) by randomly placing papers (low friction) on the pool table (high friction).

## 6.2 DATA COLLECTION

Since the low-level controller in the Sphero BOLT robot is embedded in the robot's computational board (unknown to users) and the uncertainty of the robot's dynamics is complicated, it is hard to build a simulation environment that has small sim-to-real gap to the real-world platform. Therefore, we choose to collect data directly on the real-world platform by commanding the robot with random actions from a randomly placed location on the pool table. In sum, we collect nearly 2,000 pairs of (state, action, next state) for each texture. Regarding the privileged information in Phase 1 training, we assign relatively accurate friction coefficients for each texture. Note that we do not collect data for out-of-distribution situations in which the robot crosses between the paper and pool table textures, and the dynamics learned in Phase 2 are expected to generalize to such scenarios.

## 6.3 RESULTS OF GOAL-REACHING AND PATH-TRACKING TASKS

In the goal-reaching task, each layout is tested from a random initial position with the table center as the goal. The robot must reach and hover at the goal, and performance is measured by the minimum distance between its trajectory and the goal. For the path-tracking task, we test two reference tracks: a circular path and a triangular path that provides a more aggressive reference. Each layout used a similar start position. The robot is required to follow the path, and performance is measured by the average Euclidean distance between the trajectory and the closest path point. The cost design follows $J1$ (goal reaching) and $J2$ (path tracking) in Section A.6.2. Table 4 reports results on two friction layouts, each averaged over three runs from similar start poses. Since the fixed NODE model does not adapt to environment changes, its dynamics use the friction coefficient at the start location. Results show that AD-NODE achieves smaller errors, indicating effective adaptation to spatially varying friction. It also achieves lower standard deviation, demonstrating greater robustness and higher repeatability under real-world uncertainty. AD-NODE is deployable on real-world systems and outperforms the fixed NODE, which treats environmental changes as disturbances. AD-NODE also handles surface boundary crossings more reliably, while the fixed NODE often gets stuck or loses track. The videos from the experiment are available here, and the trajectories for path tracking are available in Section A.8.

## 7 CONCLUSION & FUTURE WORKS

In this paper, we propose Adaptive Dynamics learning based on NODE (AD-NODE), a method that can be integrated into MPC to perform navigation tasks on mobile robotic systems. We adopt a two-phase training process to reconstruct environmental factors and adjust state predictions accordingly. Simulation results demonstrate the superior performance of AD-NODE on a differential wheeled robot and a quadrotor operating under out-of-distribution environmental conditions. We also demonstrate the outstanding performance of applying the framework on a real-world mobile robot. Compared to a method that does not adapt its dynamics, AD-NODE can adjust its dynamics according to the environment and thereby achieve better performance in navigation tasks. In the future, we hope to extend the framework to different robotic platforms such as quadruped robots or humanoid robots.

## 8 REPRODUCIBILITY STATEMENT

We have taken several steps to ensure reproducibility. Simulation details, including the Mujoco simulation settings for the differential wheeled robot and the equations used in the quadrotor environment, are provided in A.3. Details of the dynamics learning setup, including the model architectures for all baselines and our proposed model, as well as all training parameters of our proposed model, are provided in A.4. The data collection procedure is described in Section 5.2. Implementation details of the MPC, including controller parameters and the cost function, are included in A.6. All theoretical results are supported by complete derivations and proofs in A.1 and A.2. The source code will be released upon acceptance.

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

# A APPENDIX

## A.1 ERROR PROPAGATION IN NODE VS. DISCRETE-TIME MAP

We compare error growth when dynamics are approximated by (i) a continuous-time vector field model (NODE), and (ii) a discrete-time map model (MLP). The difference lies in how approximation errors propagate over long horizons.

**Lemma A.1** (Error propagation for vector-field models (NODE)). *Consider the true continuous-time dynamics*

$$\dot{x}(t) = f(x(t), u(t)), \quad x(0) = x_0, \tag{4}$$

*and the approximate continuous-time dynamics*

$$\dot{\hat{x}}(t) = \hat{f}(\hat{x}(t), u(t)), \quad \hat{x}(0) = x_0, \tag{5}$$

*where $f, \hat{f} : \mathbb{R}^d \times \mathcal{U} \to \mathbb{R}^d$, $f$ is Lipschitz continuous in $x$ with Lipschitz constant $L$ and let the approximation error be*

$$\epsilon = \sup_{x \in \mathbb{R}^d, \, u \in \mathcal{U}} \|f(x, u) - \hat{f}(x, u)\|. \tag{6}$$

*Based on $\delta(0) = 0$, the trajectory error $\delta(t) = \|x(t) - \hat{x}(t)\|$ satisfies the bound*

$$\delta(t) \le \frac{\epsilon}{L} \big( e^{Lt} - 1 \big). \tag{7}$$

**Proof.** *Let $e(t) = x(t) - \hat{x}(t)$. Then*

$$\dot{e}(t) = f(x(t), u(t)) - \hat{f}(\hat{x}(t), u(t)).$$

*Adding and subtracting $f(\hat{x}(t), u(t))$ gives*

$$\dot{e}(t) = \big( f(x(t), u(t)) - f(\hat{x}(t), u(t)) \big) + \big( f(\hat{x}(t), u(t)) - \hat{f}(\hat{x}(t), u(t)) \big).$$

*Taking norms and using the Lipschitz property,*

$$\|\dot{e}(t)\| \le L\|e(t)\| + \epsilon.$$

*Applying Grönwall's inequality with initial error $\|e(0)\| = 0$ yields*

$$\|e(t)\| \le \frac{\epsilon}{L}(e^{Lt} - 1).$$

□

**Lemma A.2** (Error propagation for discrete-time map models (MLP)). *Consider the true discrete-time dynamics*

$$x_{k+1} = F(x_k, u_k), \quad x_0 \in \mathbb{R}^d, \tag{8}$$

*and the approximate model*

$$\hat{x}_{k+1} = \hat{F}(\hat{x}_k, u_k), \quad \hat{x}_0 = x_0, \tag{9}$$

*where $F, \hat{F} : \mathbb{R}^d \times \mathcal{U} \to \mathbb{R}^d$, $F$ is Lipschitz continuous in $x$ with constant $L$, and the one-step approximation error is bounded by*

$$\|F(x, u) - \hat{F}(x, u)\| \le \epsilon, \quad \forall (x, u) \in \mathbb{R}^d \times \mathcal{U}. \tag{10}$$

*Based on zero initial error, after $H$ steps, the trajectory error satisfies*

$$\|x_H - \hat{x}_H\| \le \epsilon \frac{L^H - 1}{L - 1}. \tag{11}$$

***Proof.*** *Define $e_k = x_k - \hat{x}_k$. Then*

$$e_{k+1} = F(x_k, u_k) - \hat{F}(\hat{x}_k, u_k).$$

*Adding and subtracting $F(\hat{x}_k, u_k)$ gives*

$$e_{k+1} = \big(F(x_k, u_k) - F(\hat{x}_k, u_k)\big) + \big(F(\hat{x}_k, u_k) - \hat{F}(\hat{x}_k, u_k)\big).$$

*Taking norms,*

$$\|e_{k+1}\| \le L\|e_k\| + \epsilon.$$

*By induction with $e_0 = 0$, the recursive inequality solves to*

$$\|e_H\| \le \epsilon \sum_{i=0}^{H-1} L^i = \epsilon \frac{L^H - 1}{L - 1}.$$

$$\square$$

**Summary**  The theoretical results above show that NODE suffers from exponential error growth with horizon length and MLP suffers from geometrical error growth with horizon length. However, the source of error differs: NODE accumulate error through the *vector-field approximation*, while MLPs inject fresh error at every prediction step. Based on the results from Chen et al. (2018), this structural distinction implies that NODE yield smoother and more consistent rollouts under bounded model mismatch. Moreover, empirical evidence from Chen et al. (2018) demonstrates that NODE can achieve comparable or superior performance to discrete models with fewer parameters. Taken together, these results suggest that while NODE do not eliminate long-horizon error amplification, they provide practical advantages in stability, efficiency, and trajectory consistency, making them a favorable choice for model-based control.

## A.2   CONVERGENCE ANALYSIS OF MPC BASED ON LEARNED DYNAMICS ERROR

In this section, we start deriving dynamics errors $\varepsilon_f$ and analyze convergence of MPC with the learned dynamics function.

The true plant dynamics function $f$ is defined in Equation 1 and the learned dynamics model $\hat{f}$ is composed of two components:

1. Adaptive module: $\hat{z}_k = h(\{(x_i, u_i)\}_{i=k-M}^{k-1})$ encodes the past $M$ steps of observed state-action history into an adaptive latent vector $\hat{z}_k$.

2. State network: $n(x_k, u_k, \hat{z}_k)$ predicts the next state using the current state $x_k$, current control $u_k$, and the adaptive latent $\hat{z}_k$.

The overall learned-model one-step error is defined as

$$\varepsilon_f := \|f(x_k, u_k, \hat{e}_k) - f(x_k, u_k, e_k)\|,$$

which bounds the difference between the true next state and the predicted next state.

This error $\varepsilon_f$ can be decomposed into two contributions:

1. Adaptive module error: Let $\varepsilon_z := \|\hat{z}_k - z_k^\star\|$ denote the error of the adaptive module in estimating the true latent environment variables. Assuming the state network is Lipschitz in $\hat{z}$, i.e.,

$$\|n(x, u, \hat{z}_1) - n(x, u, \hat{z}_2)\| \le L_z \|\hat{z}_1 - \hat{z}_2\|,$$

then the contribution of the adaptive module to the one-step approximation error of state network is bounded by

$$\|n(x_k, u_k, \hat{z}_k) - n(x_k, u_k, z_k^\star)\| \le L_z \, \varepsilon_z.$$

2. State network approximation error: Even if the adaptive module were perfect ($\hat{z}_k = z_k^\star$), the state network $n$ may still have intrinsic approximation error:

$$\varepsilon_s := \|n(x_k, u_k, z_k^\star) - f(x_k, u_k, e_k)\|.$$

**Combined one-step error:** By the triangle inequality, the total one-step learned-model error is

$$\varepsilon_f = \|f(x_k, u_k, \hat{e}_k) - f(x_k, u_k, e_k)\| \le \underbrace{L_z \, \varepsilon_z}_{\text{adaptive module contribution}} + \underbrace{\varepsilon_s}_{\text{state network contribution}} .$$

After deriving one-step dynamics error $\varepsilon_f$, we start deriving the overall convergence of MPC using this dynamics. We consider the learned dynamics model $\hat{f}(x, u, \hat{e})$ with estimated environmental factor $\hat{e}$, and assume:

1. Lipschitz dynamics: $\|f(x_1, u, e) - f(x_2, u, e)\| \le L_x \|x_1 - x_2\|, \forall x_1, x_2, u$.
2. Bounded one-step model error: $\varepsilon_f$.
3. Lipschitz stage and terminal costs: $|\ell(x_1, u) - \ell(x_2, u)| \le L_\ell \|x_1 - x_2\|, |l_f(x_1) - l_f(x_2)| \le L_{l_f} \|x_1 - x_2\|$.
4. Existence of terminal ingredients: terminal set $\mathcal{X}_f$ and terminal control law $k_f$ ensure recursive feasibility and nominal decrease.

**Lemma A.3** (Predicted vs actual finite-horizon cost difference). *Let $\hat{U}_k^\star$ be the optimal MPC sequence at time $k$ with predicted states $\hat{x}_{k+i|k}$. Denote the true states under $\hat{U}_k^\star$ by $x_{k+i}$. Then*

$$\left| \sum_{i=0}^{N-1} \ell(x_{k+i}, \hat{u}_{k+i|k}) + l_f(x_{k+N}) - \sum_{i=0}^{N-1} \ell(\hat{x}_{k+i|k}, \hat{u}_{k+i|k}) - l_f(\hat{x}_{k+N|k}) \right| \le \Gamma_N \, \varepsilon_f, \quad (12)$$

*where*

$$\Gamma_N = L_\ell \sum_{i=0}^{N-1} \frac{1 - L_x^i}{1 - L_x} + L_{l_f} \frac{1 - L_x^N}{1 - L_x}. \quad (13)$$

*Proof. Using the one-step model error bound and Lipschitz dynamics, the state prediction error grows as*

$$\|\hat{x}_{k+i+1|k} - x_{k+i+1}\| \le L_x \|\hat{x}_{k+i|k} - x_{k+i}\| + \varepsilon_f.$$

*Then by the Lipschitz property of the stage and terminal costs,*

$$|\ell(x_{k+i}, \hat{u}_{k+i|k}) - \ell(\hat{x}_{k+i|k}, \hat{u}_{k+i|k})| \le L_\ell \|\hat{x}_{k+i|k} - x_{k+i}\|,$$

$$|l_f(x_{k+N}) - l_f(\hat{x}_{k+N|k})| \le L_{l_f} \|\hat{x}_{k+N|k} - x_{k+N}\|.$$

*Summing over $i = 0, \ldots, N-1$ gives the stated bound.*

**Lemma A.4** (Descent inequality for the MPC value function). *Let $V_k(x_k)$ denote the MPC value at time $k$. Then, under recursive feasibility,*

$$V_k(x_{k+1}) - V_k(x_k) \le -\ell(x_k, u_k) + 2\Gamma_N \varepsilon_f, \quad (14)$$

*where $u_k$ is the applied MPC control and $\Gamma_N$ is as in Lemma A.3.*

*Proof.*

1. *Construct the shifted candidate sequence for time $k + 1$: $\tilde{U}_{k+1} = \{\hat{u}_{k+1|k}, \ldots, \hat{u}_{k+N-1|k}, k_f(\hat{x}_{k+N|k})\}$.*

2. *By definition of the MPC value function, $V_{k+1}(x_{k+1}) \leq J_{k+1}^{pred}(\tilde{U}_{k+1})$.*

3. *Apply Lemma A.3 at time $k+1$: $|J_{k+1}^{pred}(\tilde{U}_{k+1}) - J_{k+1}^{true}(\tilde{U}_{k+1})| \leq \Gamma_N \varepsilon_f$.*

4. *Relate the true tail cost to the full cost: $J_{k+1}^{true}(\tilde{U}_{k+1}) = J_k^{true}(\hat{U}_k^\star) - \ell(x_k, u_k)$.*

5. *Apply Lemma A.3 at time $k$ to relate $J_k^{true}(\hat{U}_k^\star)$ to $V_k(x_k)$: $J_k^{true}(\hat{U}_k^\star) \leq V_k(x_k) + \Gamma_N \varepsilon_f$.*

6. *Combining the above steps gives*

$$V_k(x_{k+1}) - V_k(x_k) \leq -\ell(x_k, u_k) + 2\Gamma_N \varepsilon_f.$$

**Lemma A.5** (Uniformly Ultimate Boundedness). *Assume $V_k(x)$ is positive definite and satisfies $\underline{\alpha}_V(\|x - x^\star\|) \leq V_k(x) \leq \overline{\alpha}_V(\|x - x^\star\|)$. Define*

$$r = \underline{\alpha}^{-1}(2\Gamma_N \varepsilon_f), \tag{15}$$

*where $\underline{\alpha}$ is the class-$\mathcal{K}$ function in Lemma A.4. Then the closed-loop is uniformly ultimate bounded:*

$$\limsup_{k \to \infty} \|x_k - x^\star\| \leq r. \tag{16}$$

**Summary**  The overall one-step learned-model error $\varepsilon_f$, which combines contributions from the adaptive module and the state network, directly determines the practical convergence bound of the MPC. Specifically, under the robust descent inequality of Lemma A.4, the closed-loop trajectories are guaranteed to converge to a neighborhood of the equilibrium of radius

$$r = \underline{\alpha}^{-1}(2\Gamma_N \varepsilon_f),$$

where $\Gamma_N$ depends on the prediction horizon and Lipschitz constants of the stage and terminal costs, and $\underline{\alpha}$ characterizes the Lyapunov decrease. If $\underline{\alpha}$ is quadratic, this simplifies to $r = \sqrt{2\Gamma_N \varepsilon_f / \alpha_1}$, showing that the attractor radius scales as the square root of the learned-model error. Therefore, improvements in either the adaptive module (reducing $\varepsilon_z$) or the state network (reducing $\varepsilon_s$) directly shrink $\varepsilon_f$, which in turn reduces the size of the practical attractor and brings the closed-loop system closer to the true equilibrium.

## A.3 SIMULATION DETAILS

### A.3.1 2D DIFFERENTIAL WHEELED ROBOT WITH DIFFERENT SURFACE TEXTURES

Since MuJoCo physics engine (Todorov et al., 2012) provides good simulation for contact-rich scenarios, we built the differential wheeled robot environment from scratch in MuJoCo. To the best of our knowledge, existing RL environments for ground vehicle navigation are either built using a bicycle model or assume that the tire undergoes a pure rotation on the ground. In addition, they often ignore turning friction or rolling friction, oversimplifying the real-world situations of a ground vehicle that navigates on different surfaces.

To simulate real-world situations and evaluate the effectiveness of our approach, we build a differential wheeled robot with wheel torques as control input that considers slipping, turning, and rolling frictions of the surface. However, such contact-rich scenes often lead to simulation instability, and there are no constraints ensuring that the wheels remain in contact with the surface at all times during the robot's movement. Therefore, we set the damping coefficients for each joint at 0.1 Ns / m and the integrator to be a *fourth-order Runge-Kutta* method to prevent the simulation from exploding.

To simulate real-world robots, we develop a low-level PID controller that can transform the high-level control inputs, desired forward velocity $u_{\text{forward}}$ and desired steering angle $u_{\text{turn}}$, to low-level wheel torques. The implemented low-level controller is expressed as

$$u_{left}[k] = K_{p,v} \cdot e_v[k] + K_{i,v} \cdot \sum_{j=0}^{k} e_v[j] \cdot dt + K_{p,h} \cdot u_{\text{turn}}[k] + K_{d,h} \cdot \frac{u_{\text{turn}}[k] - u_{\text{turn}}[k-1]}{dt}$$

$$u_{right}[k] = K_{p,v} \cdot e_v[k] + K_{i,v} \cdot \sum_{j=0}^{k} e_v[j] \cdot dt - K_{p,h} \cdot u_{\text{turn}}[k] - K_{d,h} \cdot \frac{u_{\text{turn}}[k] - u_{\text{turn}}[k-1]}{dt}$$

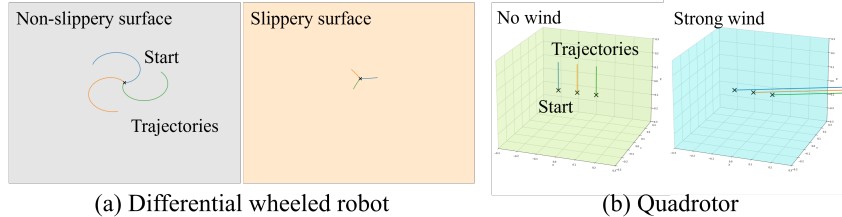

(a) Differential wheeled robot  (b) Quadrotor

Figure 6: The influence of environmental factors on differential wheeled robots and quadrotors under identical control inputs: (a) It is more difficult for a differential wheeled robot to navigate on a slippery surface compared to a non-slippery one. (b) Even when thrust is applied to a quadrotor, strong wind can significantly alter its dynamics, causing it to drift in the x-direction.

where $e_v[k] = u_{\text{forward}} - v[k]$, $u_{left}$ denotes left wheel torque, $u_{right}$ denotes right wheel torque, and $dt$ denotes the control time step, which is set to be 0.04 seconds. $K_{p,v}$ is the velocity proportional control gain, $K_{i,v}$ is velocity integral control gain, $K_{p,h}$ is the heading proportional control gain, and $K_{d,h}$ is the heading derivative control gain.

However, the simulated differential wheeled robots often have jerky motions, but we still collect data and test our framework in the environment to see whether our framework can handle complex dynamics. Figure 6 shows the resulting trajectories on different surfaces with control inputs of $[0.5, 0.5]^T$ applied for 80 steps. The surface conditions used to collect data are $\mu_{\text{sliding}}, \mu_{\text{turning}}, \mu_{\text{rolling}}$ = [0.7, 0.04, 0.01] for the slippery surface and [2, 0.005, 0] for the non-slippery surface. Figure 6 demonstrate that surface friction significantly affects the dynamic behavior of a differential wheeled robot. On a slippery surface, the robot travels a shorter distance and turns less under the same control actions, highlighting the importance of dynamics adjustment for achieving a robust controller.

### A.3.2   3D QUADROTOR WITH DIFFERENT WIND FIELDS

In the quadrotor platform, we consider the following quadrotor dynamics as our governing equations in the simulator, which also follows the implementation from Huang et al. (2023).

$$\dot{p} = v, \qquad\qquad m\dot{v} = mg + Re_3 f_\Sigma + d,$$
$$\dot{R} = RS(\omega), \qquad\qquad J\dot{\omega} = J\omega \times \omega + \tau,$$

where $p, v, g \in \mathbb{R}^3$ are the position, velocity, and gravity vectors in the world frame, $R \in \text{SO}(3)$ is the attitude rotation matrix, and $\omega \in \mathbb{R}^3$ is the angular velocity in the body frame. The parameters $m$ and $J$ represent the mass and inertia matrix, respectively. The unit vector $e_3 = [0; 0; 1]$, and $S(\cdot) : \mathbb{R}^3 \to \mathfrak{so}(3)$ maps a vector to its skew-symmetric matrix form.

Regarding the environmental factors, $d$ is the translational disturbance, which models the wind fields. The control inputs are the total thrust $f_\Sigma$ and the torque $\tau$ in the body frame. For quadrotors, there exists a linear and invertible actuation matrix between $[f_\Sigma; \tau]$ and the four motor speeds. Figure 6 shows the trajectories under different wind fields with a 0.1 N thrust applied for 50 steps. The strong wind field introduces a 0.5 N disturbance in the positive x-direction. The resulting trajectories show that the wind significantly alters the dynamics, posing challenges for the original MPC, which does not adapt its model of the system dynamics.

### A.4   IMPLEMENTATION DETAILS OF DYNAMICS LEARNING

### A.4.1   MODEL STRUCTURES

We implement the backbone of the state net using an MLP. For the differential wheeled robot, it consists of two hidden fully connected layers, each with 64 units followed by ReLU activation. For the quadrotor, it has three hidden fully connected layers, each with 64 units followed by ReLU activation. The method is implemented with Torchdiffeq package (Chen, 2018), and *Euler integrator* is adopted as the ODE solver.

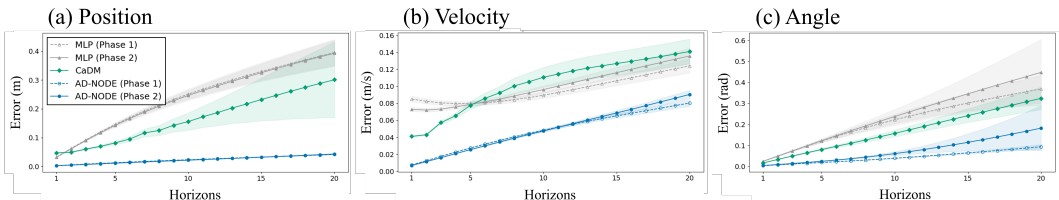

Figure 7: Dynamics prediction errors (RMSE) for the differential wheeled robot on the test set over different horizons: (a) position, (b) velocity, and (c) angle.

In the differential wheeled robot environment, both the environmental encoder and the adaptive module are implemented using an MLP consisting of one hidden fully connected layer with 64 units, followed by ReLU activation. For the quadrotor, the environmental encoder is also implemented using an MLP with one hidden fully connected layer and 64 units, followed by ReLU activation. The adaptive module is implemented using a 1D CNN consisting of three stacked 1D convolutional layers (with kernel sizes of 5 and 3), each followed by ReLU activation and dropout. The extracted features are then flattened and passed through an MLP comprising two linear layers with ReLU activation and dropout.

### A.4.2 HYPERPARAMETER

In both Phase 1 training and Phase 2 training, we use Adam as an optimizer and MSE as loss function to train the networks. However, there are some differences in terms of training hyperparameters between each phase and each environment.

For the differential wheeled robot environment, in Phase 1 training, a learning rate of $1 \times 10^{-3}$ is used and decayed to $1 \times 10^{-4}$, with a batch size of 512. In Phase 2 training, the learning rate is set to $1 \times 10^{-3}$, and the adaptive module is trained with a batch size of 128. The experiments of all models were trained on a single NVIDIA RTX 3070 GPU on a personal workstation. Training time for each experiment ranged from 1 to 5 hours, depending on model complexity and dataset size.

For the quadrotor environment, we apply curriculum learning during Phase 1 training. An exponential learning rate scheduler is used, starting from a learning rate of $1 \times 10^{-3}$ and decaying to $1 \times 10^{-4}$. The training starts from learning to match 1 future step to 30 future steps. In each curriculum, we train 10 epochs with 1024 batch size. Similar to the experiment done on the differential wheeled robot platform, the training process is conducted on a single NVIDIA RTX 3090 GPU, the runtime is around three hours. Regarding Phase 2 training, we set the learning rate to be $1 \times 10^{-4}$ and train 100 epoch to learn the adaptive module with 1024 batch size. The entire training time on a single NVIDIA RTX 3090 GPU is around one hour.

### A.4.3 ADDITIONAL EVALUATION OF DYNAMICS MODEL PERFORMANCE

**2D Differential Wheeled Robot** We collect a test dataset using the same collection procedure with a sampling parameter that differs from the one used during training data collection to evaluate the long-horizon prediction error of a learned dynamics. Then, we compare each dynamics learning method in terms of errors in position, velocity and heading angle at each prediction time up until 20 horizons. The errors are defined as RMSE between the prediction and the ground-truth.

Figure 7 shows the prediction errors of each dynamics learning method on the test dataset. Compared to baselines such as CaDM and MLP, AD-NODE achieves lower prediction errors across all state components. The results also align with the theoretical conclusion from A.1. Learning-based dynamics models are typically prone to error accumulation over time, but our method mitigates this issue by using NODE as the backbone, which better captures the continuous-time evolution of robot dynamics. Models with access to privileged information achieve higher accuracy than those relying solely on inferred context from historical state-action trajectories.

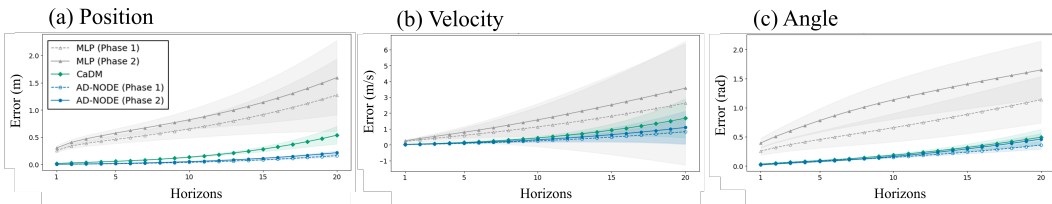

Figure 8: Dynamics prediction errors (RMSE) for the quadrotor on the test set over different horizons: (a) position, (b) velocity, and (c) angle.

**3D Quadrotor**   We evaluate the dynamics model using the same procedure as for the differential wheeled robot. However, since orientation lies in SO(3), orientation error is defined as the minimal rotation angle between the predicted and ground-truth quaternions. Given a ground-truth quaternion $q_{gt}$ and a predicted quaternion $\hat{q}$, the error quaternion $q_e$ is defined as $q_e = q_{gt} \otimes \hat{q}^{-1}$ where $\otimes$ denotes the quaternion product. The angle of the shortest rotation from $\hat{q}$ to $q_{gt}$ can be computed as $2 \cdot \arccos\left(|q_e^w|\right)$, where $q_e^w$ is the scalar (real) part of the quaternion $q_e$.

Figure 8 presents the prediction error of each dynamics learning method on the test dataset. Similar to the differential wheeled robot environment, our proposed dynamics model demonstrates lower long-horizon prediction errors compared to the baselines. The accumulated error is particularly evident in the MLP-based model, which fails to capture the continuity inherent in physical dynamics. CaDM performs better than MLP due to its use of forward and backward loss.

### A.5   Implementation Details of Baselines

#### A.5.1   Model Structures

In this paper, we implement four baselines across both environments: (1) MLP-based dynamics with privileged information, (2) MLP-based dynamics with historical information, (3) CaDM from Lee et al. (2020), and (4) the meta-learning-based dynamics model from Belkhale et al. (2021). Below, we describe the model architectures for these baselines.

The same model structures are used in both simulation environments. For the MLP-based dynamics models (with privileged and historical information), the main network is implemented as an MLP with three hidden fully connected layers (64 units, ReLU activation). The encoder is implemented as an MLP with two hidden fully connected layers, also with 64 units and ReLU activation.

For CaDM, the dynamics model consists of an MLP with two hidden fully connected layers (128 units, ReLU activation) for both the forward and backward dynamics. The encoder is implemented as an MLP with one hidden layer (64 units, ReLU activation).

For the meta-learning–based dynamics model, the context encoder is a variational encoder that maps a history of state-action pairs to a latent context vector. It consists of a two fully connected hidden layers (64 units, ReLU activation) shared between the mean and log-variance outputs, producing a Gaussian distribution from which the latent vector can be sampled. The forward dynamics model is an MLP with two hidden layers (128 units, ReLU activation). During training, the model minimizes MSE loss between predicted and true states and the Kullback-Leibler divergence (KL divergence) between the inferred context distribution and a standard Gaussian.

#### A.5.2   Hyperparameter

The Adam optimizer is used for all baselines. For the MLP-based dynamics model with privileged information, the number of training epochs is 100, with a learning rate of $1 \times 10^{-4}$ and a batch size of 128. In the Phase 2 counterpart of the MLP-based dynamics model, where the encoder is retrained using historical information, training is conducted for 15 epochs with a learning rate of $5 \times 10^{-5}$ and a batch size of 128. For CaDM, a batch size of 256, a learning rate of $1 \times 10^{-4}$, and 30 training epochs are used. When computing the loss, the weight ratio of the backward to the forward models is set to 0.5. For the meta-learning-based method, a batch size of 256, a learning

rate of $1 \times 10^{-4}$, and 50 training epochs are used. When computing the loss, the weight ratio of the KL divergence term to the forward loss is set to $1 \times 10^{-2}$.

## A.6 IMPLEMENTATION DETAILS OF MPC

### A.6.1 HYPERPARAMETER

Hyperparameters of the MPPI controller vary based on the task, cost function design and dynamics function. For the differential wheeled robot environment, in the goal-reaching task, the horizon is set to 20, the number of samples to 500, and the temperature to $1 \times 10^{-2}$. The sampling standard deviation for each control dimension is $[0.1, 0.1]^T$ corresponding to $u_{\text{forward}}$ and $u_{\text{turn}}$, respectively. In the path tracking task, the horizon is set to 15, the number of samples to 800, and the temperature to $1 \times 10^{-4}$. The sampling standard deviation for each control dimension is $[0.5, 0.3]^T$ corresponding to $u_{\text{forward}}$ and $u_{\text{turn}}$. In the velocity tracking task, the horizon is set to 20, the number of samples to 800, and the temperature to $1 \times 10^{-4}$. The sampling standard deviation for each control dimension is $[0.2, 0.1]^T$ corresponding to $u_{\text{forward}}$ and $u_{\text{turn}}$. The length of the state-action history is set to 5 for all tasks.

For the quadrotor environment, horizon is set at 40, number of sampling size is set at 4096, temperature is set at 0.05, sampling standard deviation for each control dimension is $[0.25, 0.7, 0.7, 0.7]^T$, each corresponds to thrust, angular velocity in raw, pitch and yaw direction. The length of the state-action history is set at 10.

During inference, the length of the state-action history is shorter than the designated length in the beginning of each episode. Therefore, we apply random actions at the beginning to fill the designated length. Since the robot does not know the ground-truth environmental factors, this is a way for it to capture the current environment by randomly exploring for a short time at the start of each episode. Compared to the total length of the controller, which usually operates over hundreds or thousands of steps, this initial exploration does not sacrifice the accuracy or success rate too much.

### A.6.2 COST DESIGN

For the differential wheeled robot and the real-world deployment, we design two cost functions: $J_1$ for goal reaching and $J_2$ for path tracking. The two definitions are

$$J_1 = \sum_{k=0}^{H} \left[ w_v \left( v_k - v_k^{\text{ref}} \right)^2 + w_\theta \left( \theta_k - \theta_k^{\text{pp}} \right)^2 \right],$$

$$J_2 = \sum_{k=0}^{H} \left[ w_p \left\| p_k - p_k^{\text{ref}} \right\|^2 + w_v \left( v_k - v_k^{\text{ref}} \right)^2 + w_\theta \left( \theta_k - \theta_k^{\text{ref}} \right)^2 \right],$$

where

$$p_k = \begin{bmatrix} x_k \\ y_k \end{bmatrix}, \quad p_k^{\text{ref}} \text{ is the reference position,}$$

$$v_k = \left\| \begin{bmatrix} \dot{x}_k \\ \dot{y}_k \end{bmatrix} \right\|_2, \quad v_k^{\text{ref}} \text{ is the reference velocity,}$$

$$\theta_k^{\text{pp}} = \arctan 2 \left( y_{\text{goal}} - y_k, \, x_{\text{goal}} - x_k \right),$$

$$\theta_k^{\text{ref}} \text{ is the reference heading,}$$

and $w_p, w_\theta, w_v$ are the positive scalar weights on position, heading, and velocity losses respectively.

The MPC cost function for a quadrotor performing a goal reaching and hovering or path tracking task is defined as

---

**Algorithm 1** Online Dynamics Learning

---

1: **Initialize:** dataset $\mathcal{D} \leftarrow \emptyset$, State Net parameters $\theta$ from Phase 1, Adaptive Module parameters $\phi$, environment $\mathcal{E}$, environmental factors $e$, MPC controller $\pi_{\text{MPC}}$
2: **for** episode $= 1$ to $N_{\text{episodes}}$ **do**
3:     $x_1 \leftarrow \mathcal{E}.\text{reset}()$
4:     **for** timestep $k = 1$ to $T$ **do**
5:         **if** explore(episode, $k$) **then**
6:             $u_k \leftarrow$ sample random action from action space
7:         **else**
8:             $u_k \leftarrow \pi_{\text{MPC}}(x_k)$           ▷ Plan using current Adaptive Module and State Net
9:         **end if**
10:      $x_{k+1}, r \leftarrow \mathcal{E}.\text{step}(u_k)$
11:      $\mathcal{D} \leftarrow \mathcal{D} \cup \{(\{(x_i, u_i)\}_{i=k-M}^{k-1}, e_k)\}$
12:      **if** model update condition met **then**
13:         $\phi \leftarrow$ update Adaptive Module using data $\mathcal{D}$
14:      **end if**
15:      $x_k \leftarrow x_{k+1}$
16:     **end for**
17: **end for**

---

$$J_3 = \sum_{k=0}^{H} \left( w_p \left\| p_k - p_k^{\text{ref}} \right\|^2 + w_q \left( 1 - \left( q_k^\top q_k^{\text{ref}} \right)^2 \right) \right),$$

where

$$p_k^{\text{ref}} \text{ is the reference (goal or trajectory) position at time } k,$$

$$q_k^{\text{ref}} \text{ is the reference quaternion at time } k,$$

and $w_p, w_q$ are weighting factors for the position and orientation errors, respectively.

### A.6.3 ONLINE DYNAMICS LEARNING

Online dynamics learning follows the general procedure in model-based RL. The pseudo code of online dynamics learning is shown at Algorithm 1.

Online learning is used in the quadrotor task, as described in Section 4.2, to refine our dynamics model. To further evaluate the effectiveness of online learning under extreme conditions, we conduct an experiment comparing performance before and after online adaptation. In scenarios where wind forces exceed 3 N, which is far beyond the offline training range of −1 to 1 N, offline learning yielded a position RMSE of $0.1649\ m \pm 0.0221$, while online finetuning reduced it to $0.0646\ m \pm 0.0153$. The results demonstrate that online learning can significantly improve performance in new environments by refining the dynamics model using rollout trajectories.

### A.7 PERFORMANCE OF THREE-DIMENSIONAL WIND FIELD

We evaluate both model-free approaches and model-based approaches on three-dimensional wind field with random time-varying noise modeled as Brownian motion (shown in Figure 9). The performance persists even on this complex wind field. The reason why AD-NODE (Phase 1) performs worse than AD-NODE (Phase 2) is because Phase 1 model uses privilege information that is far from training distribution, which might obtain a wrong environmental vector. Whereas, Phase 2 model uses state-action history to reconstruct environmental vector, demonstrating strong generalization ability to unseen environmental factors.

Table 5: Performance of the quadrotor is evaluated under three dimensional wind field with random time-varying noise modeled as Brownian motion. Shown as RMSE (m).

|  | Goal-reaching | Path-tracking |
|---|---|---|
| MLP(Phase 1) | > 0.2 | > 0.2 |
| MLP(Phase 2) | > 0.2 | > 0.2 |
| CaDM | > 0.2 | > 0.2 |
| Meta-learning based | 0.088 ± 0.021 | 0.109 ± 0.075 |
| Fixed NODE | 0.066 ± 0.012 | 0.061 ± 0.025 |
| RMA | 0.069 ± 0.019 | 0.063 ± 0.018 |
| DATT | 0.056 ± 0.016 | 0.050 ± 0.015 |
| AD-NODE(Phase 1) | 0.033 ± 0.022 | 0.071 ± 0.056 |
| AD-NODE(Phase 2) | **0.021 ± 0.013** | **0.039 ± 0.023** |

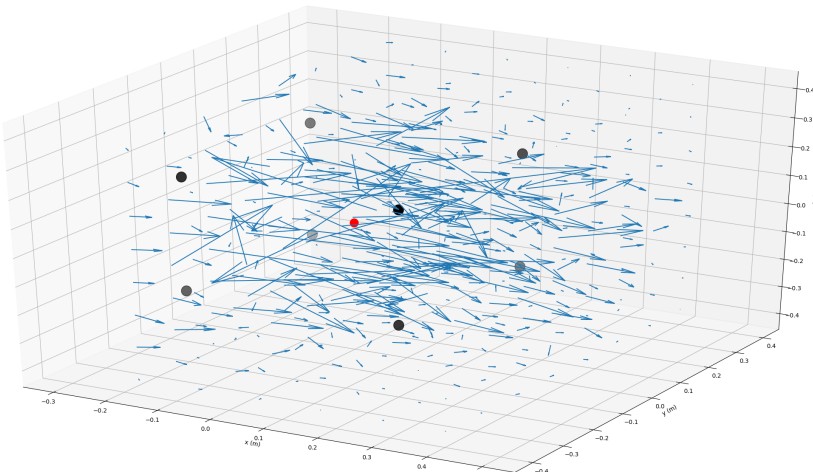

Figure 9: The figure shows three-dimensional wind field used for evaluating each algorithm. Blue arrow denotes the vector field of wind force. Black dot is the start and red dot is the goal.

## A.8 TRAJECTORIES OF THE PATH-TRACKING TASKS

### A.8.1 TRAJECTORIES OF MODEL-FREE BASELINES

In this section, we provide a qualitative analysis of the trajectories generated by our proposed model and the model-free baselines for the path-tracking task when combined with MPPI control. Figure 10 shows the trajectories in the quadrotor simulator. We observe that the quadrotor using RMA and DATT as policies can reach positions near the target path. However, they fail to track the path accurately, resulting in higher errors compared to AD-NODE. These results align with those shown in Table 3.

### A.8.2 TRAJECTORIES OF REAL-WORLD EXPERIMENTS

Figure 11 shows the trajectories from the real-world experiments for the path-tracking task using circular and triangular reference paths. The trajectories appear wavy because the robot wobbles during motion, indicating high system uncertainty. This is because the robot is not a pure unicycle model, as it has a plastic shell over a differential-drive mechanism, which makes the system challenging to control. Despite this, our model still aligns more closely with the reference path compared to the fixed-dynamics model, demonstrating that AD-NODE can be deployed on systems with high uncertainty and help the robot adapt to different environments.

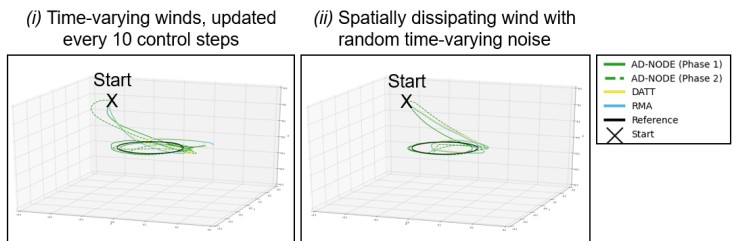

Figure 10: Trajectories of AD-NODE and model-free baselines in quadrotor simulation for path tracking.

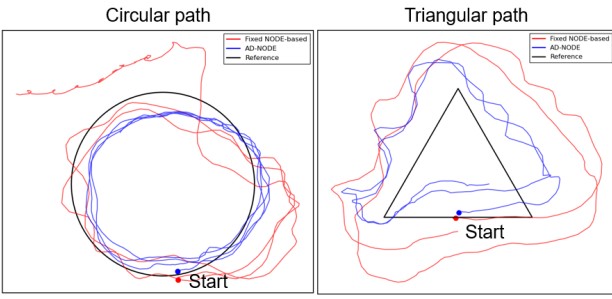

Figure 11: Trajectories of AD-NODE and fixed NODE-based dynamics for path tracking in the Sphero BOLT robot.

### A.9 Additional evaluation in environments with piecewise-constant factors

#### A.9.1 2D Differential Wheeled Robot with Different Surface Textures

For this experiment, we used the same setup and data collection method described in Section 5. Both goal-reaching and path-tracking tasks were performed in environments with piecewise-constant friction, introducing drastic changes at the crossing boundaries. Examples of the test environment are shown in Figures 1 and 6, and the results are summarized in Table 6. From the table, we can see that the benefits of using AD-NODE persist in this test case.

#### A.9.2 3D Quadrotor with Different Wind Fields

Similar to differential wheeled task, we used the same setup and data collection method described in Section 5. Both goal-reaching and path-tracking tasks were performed in environments with piecewise-constant wind fields, introducing drastic changes at the crossing boundaries. The test environments are shown in Figures 1 and 6, and the results are summarized in Table 7. From the table, we can see that the benefits of using AD-NODE persist in this test case.

Table 6: Performance of differential wheeled robot with a piecewise-constant spatial friction layouts that remain fixed over time.

|  | **Goal-reaching** | **Path-tracking** |
|---|---|---|
|  | Success rate (%) | RMSE (m) |
| MLP (Phase 1) | 2 | $> 0.1$ |
| MLP (Phase 2) | 2 | $> 0.1$ |
| CaDM | 28 | $> 0.1$ |
| Meta-learning based | 12 | $0.043 \pm 0.031$ |
| Fixed NODE-based | 32 | $0.046 \pm 0.015$ |
| AD-NODE (Phase 1) | **94** | **$0.017 \pm 0.007$** |
| AD-NODE (Phase 2) | **94** | $0.021 \pm 0.013$ |

Table 7: Performance of the quadrotor with piecewise-constant wind field layouts that remain fixed over time.

| | Goal-reaching & hovering RMSE (m) | Path-tracking RMSE (m) |
|---|---|---|
| MLP (Phase 1) | $> 0.2$ | $> 0.2$ |
| MLP (Phase 2) | $> 0.2$ | $> 0.2$ |
| CaDM | $> 0.2$ | $> 0.2$ |
| Meta-learning based | $0.052 \pm 0.041$ | $> 0.2$ |
| Fixed NODE-based | $0.099 \pm 0.058$ | $0.122 \pm 0.037$ |
| AD-NODE (Phase 1) | $\mathbf{0.013 \pm 0.008}$ | $\mathbf{0.081 \pm 0.030}$ |
| AD-NODE (Phase 2) | $0.033 \pm 0.022$ | $0.103 \pm 0.047$ |

Table 8: AD-NODE performance on test cases with different distances from the training distribution. The RMSE and standard deviation of position and velocity errors for the three categories are reported after the mass reaches the target. In the fixed-mass setting, the mass remains constant throughout the task. In the changing-mass setting, the mass starts at 5 kg and transitions to the value corresponding to each regime after 150 control steps.

| | Fixed-mass | | | Changing-mass | | |
|---|---|---|---|---|---|---|
| | Standard | Moderate | Extreme | Standard | Moderate | Extreme |
| Position RMSE (mm) | $5.600 \pm 0.524$ | $5.500 \pm 0.609$ | $5.819 \pm 1.106$ | $5.682 \pm 0.399$ | $5.993 \pm 0.734$ | $5.884 \pm 2.231$ |
| Velocity RMSE (mm/s) | $2.497 \pm 0.746$ | $3.890 \pm 2.136$ | $5.480 \pm 3.675$ | $2.131 \pm 0.378$ | $2.408 \pm 0.581$ | $3.668 \pm 1.792$ |

## A.10 ADDITIONAL EVALUATION ON MASS-SPRING-DAMPING SYSTEM

In this experiment, we want to test our algorithm on a toy example with known dynamics to understand the generalization ability of the proposed model. We consider a mass moving along a one-dimensional axis under an external force, connected to a spring and a damper. The system dynamics are described by the state-space equation as shown below:

$$\dot{x}(t) = Ax(t) + Bu(t)$$

where

$$x(t) = \begin{bmatrix} x_1(t) \\ x_2(t) \end{bmatrix}, \quad A = \begin{bmatrix} 0 & 1 \\ -\frac{k}{m} & -\frac{b}{m} \end{bmatrix}, \quad B = \begin{bmatrix} 0 \\ \frac{1}{m} \end{bmatrix}, \quad u(t) = F(t)$$

$x_1(t)$ represents the displacement and $x_2(t)$ the velocity of the mass. The environmental variation here is the changing mass during movement. Generalization performance is evaluated across three regimes in the test set: standard, moderate, and extreme. The standard regime uses masses within the training range, the moderate regime includes slightly out-of-range values, and the extreme regime includes the most distant values, similar to the definition in Lee et al. (2020). We evaluate our proposed dynamics model on a goal-reaching task, where the mass is controlled to reach a target location from various initial positions by applying an external force. Performance is reported as errors of position and velocity after the mass reaches the goal, averaged over 20 runs per category. The goal-reaching criterion requires the mass to have a position error of less than 10 mm and a velocity error of less than 10 mm/s. The results are presented in Table 8. The positional errors are comparable across categories, indicating that the model generalizes well. The fluctuations in position and velocity of the mass in the standard test samples are smaller than those observed in the extreme cases when the mass is near the target.

## A.11 THE USE OF LARGE LANGUAGE MODELS

Large language models (LLMs) were used to improve the readability of this submission. Specifically, they assisted in correcting grammar, refining sentence structure, and polishing the wording of the text. In addition, we took inspiration from LLMs and decided to use Grönwall's inequality to derive the results of Lemma A.1.

