# OpenReview forum: "AD-NODE: Adaptive Dynamics Learning with Neural ODEs for Mobile Robots Control"
_ICLR.cc/2026/Conference — Submitted to ICLR 2026_

### Official Review · Reviewer_igpy · 2025-10-15

**Soundness:** 4
**Presentation:** 3
**Contribution:** 2
**Rating:** 4
**Confidence:** 4

**Summary:**

In the paper, "AD-NODE: Adaptive dynamics learning with neural ODEs for mobile robots control," the authors propose an approach to learning adaptive dynamics models parameterized by neural ODEs that can be used in model-based planners such as MPC. The 2-stage approach first learns a latent representation of the environment from priviliged information, and then trains an adaptive module that estimates the same latent vector from state histories. The adaptive module can then be used at runtime to estimate the latent representation vector to tune the dynamics model to the environment.

Overall, I think this paper has potential, but I would like to see a stronger justification for the proposed approach and clearer contrasts with existing work. The experimental section could also benefit from more qualitative descriptions of the results. With these improvements, I believe the paper could make a solid contribution.

**Strengths:**

- The problem of adaptation for robotics and autonomous systems is timely and relevant. The approach is interesting, and could certainly be useful for making robots that can adapt to different environments at runtime. I would like to see a little more about the practical robotics side, such as how much training data is needed to work effectively, how many different environments and parameter variations are needed to train a good latent representation, how quickly the model can adapt on on-board hardware, etc.

- The experiments and baseline comparisons are comprehensive and (mostly) convincing. Barring my comments regarding the interpretability of some of the results, the experiments and baseline comparisons do a good job of evaluating the proposed approach for the task of adapting to different environments.

- The theoretical contribution of the paper appears solid.

- The paper is (mostly) well-written. There are a few areas where grammatical tools could help improve the readability or flow of the paper. For instance, even the title "AD-NODE: Adaptive Dynamics Learning with Neural ODEs for Mobile Robots Control" could be improved to "AD-NODE: Adaptive Dynamics Learning with Neural ODEs for Mobile Robot Control" (removing the plural on "robots" to sound more natural).

**Weaknesses:**

- My biggest comment is that the authors have not done enough to compare/contrast their work with RMA. RMA takes a very similar approach, but the learned object is a policy network instead of a neural ODE. The 2-stage training procedure and adaptation approach are otherwise virtually identical. What is novel about this work when compared to RMA in terms of architecture or approach? What benefit do we get aside from being model-based? Without such a comparison, the theoretical and algorithmic contribution appears minor.

  For instance, one might argue that the closed-loop policy performance of RMA confers significant benefits over the proposed approach in this paper by not needing to simulate rollouts and optimize on the fly, which can be a significant online computational burden.

  Why not compare the downstream performance of the proposed model-based method with the existing model-free RMA approach? This seems like the clearest baseline from existing work. What performance benefits are there versus the RL-based policy?

- The contrasts with related work do not clearly spell out how the proposed approach overcomes the issues faced by existing methods. The contrasts point out some common criticism of other existing approaches, such as excessive training data or limited OOD generalization, but the authors do not clearly explain how their approach overcomes these issues. As it stands, this makes the paper feel less well-motivated. What gap or limitation is the proposed approach fixing?

  The authors contrast with existing work (Kumar 2021, Zhang 2023) by saying that these approaches require large amounts of interaction data and demonstrate low sample efficiency, but do not say how their approach overcomes this downside. Simply changing to a model-based method does not alleviate this issue, and may make things worse for MPC/MPPI if the resulting model has significant model mismatch.

  The same holds true for the other related works. The authors say that meta-learning struggles with out-of-distribution generalization (this is a strong claim), but wouldn't the proposed method suffer from the same downside? It's not clear how (or if) the proposed approach handles new, unseen environments not encountered during training, whereas meta-learning approaches like MAML can theoretically handle truly novel tasks or environments since they fine-tune online.

  Without a clear contrast that demonstrates the benefits of the proposed approach over existing work, it is difficult to see exactly what advantages this approach has over existing work. The paper would significantly benefit from a clearer justification and expanded discussion.

  As a minor comment, the related work also neglects some work in robotics for adaptation to novel environments/terrains. Expanding on this could strengthen the paper to demonstrate a clearer need for the proposed approach.

- The experimental section in the main paper could benefit greatly from a more qualitative description of the performance of the controller versus the baselines. As it stands, it is difficult to determine the relative performance of the controller in the different setups.

  How do the final trajectories look for the different models? Is there a noticeable difference or does MPPI handle the mismatch?

  How quickly do the dynamics adapt to the new terrain? Is it possible to clearly distinguish the dynamics on the different surfaces? This is currently lacking in the experimental section.

**Questions:**

My questions are mostly related to the weaknesses above and are the biggest points I would like see addressed:

1. How does the proposed approach compare with RMA? The two approaches appear very similar, except that RMA learns a policy network instead of a dynamics model for MPC. What are the benefits of the proposed approach over RMA? Why not compare the downstream performance of the proposed method with RMA, or at least discuss the comparative differences between them?

2. The contrasts with related work do not clearly spell out how the proposed approach overcomes the issues faced by existing methods. What gap or limitation is the proposed approach fixing? How does the proposed approach overcome the downsides of existing work?

3. The experimental section in the main paper could benefit greatly from a more qualitative description of the performance of the controller versus the baselines. How do the final trajectories look for the different models? Is there a noticeable difference or does MPPI handle the mismatch? How quickly do the dynamics adapt to the new terrain? Is it possible to clearly distinguish the dynamics on the different surfaces? This is currently lacking in the experimental section.

---

> ### Author Response · Authors · 2025-12-03
>
> **Comparison with RMA**
>
> We appreciate the reviewer’s comment. We have included a comparison with RMA and another model-free RL method as baselines in our quadrotor simulator. Please see the general response for the results and discussion. These comparisons have also been added to the updated manuscript. We hope that the newly added results clarify the motivations and intended contributions of our approach.
>
> **Clarifying the contribution and discussion on related works**
>
> We thank the reviewer for the thoughtful comment and would like to clarify the motivation behind our model-based design choices, how our method connects to and extends prior work, and why we expect improved generalization under both offline and online settings.
>
> **Discussion of model-based approach**
> For the discussion comparing model-free and model-based approaches, please refer to the general response.
>
> **How our model bridges the gap with prior works**
> A core challenge is that many environmental variations are only partially observable through onboard sensors, leaving critical factors hidden and requiring inference from limited histories. Although many adaptive dynamics-learning models have been explored, most implicitly infer environmental factors and directly map state–action histories to next states in an end-to-end manner. Learning such mappings is difficult due to the high dimensionality of the histories and the challenge of ensuring the model captures the true environmental factors. To address this challenge, we decompose the mapping from histories to next states. This enables us to first learn environment-independent dynamics and then use simulation data to guide and to separately capture environment-specific variations. A second challenge lies in long-horizon planning, where small modeling errors quickly accumulate. Since rigid-body robot motion follows ODEs, incorporating an ODE solver into the learning process is natural. Neural ODEs learn first-order derivatives and compute system states via numerical integration, which smooths the dynamics, imposes useful structural priors, and has shown strong performance in long-term prediction tasks. Please see Section A.4.3 for dynamics model performance. We have also updated the manuscript to include this discussion.
>
> **Concerns regarding OOD**
> Regarding out-of-distribution generalization, we clarify that our model does not eliminate OOD issues, but it is expected to be more robust and data-efficient for several reasons. By explicitly leveraging environmental factors rather than relying solely on high-dimensional histories, our model better captures the underlying structure and is less prone to overfitting noise. This also enables generalization to environments that interpolate between previously seen environmental conditions. In contrast, models that take state–action histories directly as input often need to learn in a significantly larger space, making them more likely to encounter OOD scenarios. Moreover, the neural ODE formulation implicitly smooths the dynamics and adds structure, further reducing data requirements. In the online fine-tuning setting, the model additionally observes data from previously unseen environments. As shown in Section A.6.3, this allows generalization beyond the offline training distribution.
>
> **Adding more qualitative results**
>
> We thank the reviewer for the questions. We have included the qualitative description of the performance of the controller versus the baselines in our experimental section.
>
> **How quickly do the dynamics adapt to the new terrain**
>
> The length of the historical state-action sequence influences how quickly the dynamics adjust to new terrain. In our simulation environments, we generally observe adaptation within 5 time steps for the differential-drive robot and around 10 time steps for the quadrotor. In the real-world experiment, it is around 4 time steps.
>
> **Is it possible to clearly distinguish the dynamics on the different surfaces**
>
> The environmental variations are large, and visualizing the effect of applying the same action in different environments shows noticeable differences in the resulting trajectories. These trajectories can be seen in Section A.3. Additionally, in all the experiments we conducted, we included the fixed-dynamics baseline to show that the environmental changes are significant and cannot be fully handled by the robustness of MPPI, thereby justifying the need for adaptive dynamics.
>
> **Additional training information**
>
> For offline training, for the differential wheeled robot, we use 178,125 trajectories for training and 9,375 trajectories for testing (all with 20-step horizons). For the quadrotor task, we use 17,510,400 trajectories for training and 921,600 trajectories for testing (all with 20-step horizons). This dataset is collected in simulation; therefore, we believe it does not pose a significant issue.

---

### Official Review · Reviewer_zJvQ · 2025-10-26

**Soundness:** 3
**Presentation:** 3
**Contribution:** 2
**Rating:** 4
**Confidence:** 4

**Summary:**

An ode-based two-phase training procedure is used to learn latent environment representations for improving system dynamic tracking performance. In Phase 1, privileged environmental information ek is used to encoder latent dynamics zk. Next, the historical data is employed to reconstruct the latent dynamics. Lastly, the further motion state is predicted via state net module. Several tests are arranged to preliminarily validate the effectiveness.
Questions:
(1)	Although data-based methods can be very effective, the quality of their results depends on the quality of the data used for identification. In this case, the data used came from a system like quadrotor with a specific closed-loop control parameters, and is therefore only effective under these conditions. If the gains change (for example, during adjustments from the ground station), does the offline training model need to be re-obtained? The generalization in unseen environments should be provided in the manuscript.
[1] Feedback favors the generalization of neural ODEs. ICLR
[2] Millimeter-level pick and peg-in-hole task achieved by aerial manipulator. IEEE TRO
(2)	Experiments. 1) In 2D wheel robot: overly simplifies the friction settings for the movement of the vehicle in different textured environments. 2) In 3D quadrotor: real wind disturbance is three-dimensional and time-varying. It is not sufficient to consider only the constant x-direction in simulation. It is better to provide details in real fight tests. 3) The purpose of the algorithm is to improve the dynamic tracking performance by accurately predicting the future motion state. It is not sufficient to merely present the results in a table. Providing more experiment details, such as pictures and more aggressive references.
(3)	Comparison: RL-based approaches are frequently mentioned, but it has not been compared in the tests. Moreover, it is also necessary to compare some classic approximate methods including adaptive NN, disturbances observer.
[3] Precise end-effector control for an aerial manipulator under composite disturbances: Theory and experiments. IEEE TASE
(4)	In section 4.1, the authors indicate that it is effective to align two domains together in latent space even historical data carries distinct physical meanings and have significantly different dimensionality compared to ek.  Although a similar approach is proven in image field, I have some doubts about this during the process of learning dynamics. Provide more details.
[4] Image style transfer using convolutional neural networks. CVPR
(5) An ode-based two-phase training procedure is used to learn latent environment representations for improving system dynamic tracking performance. In Phase 1, privileged environmental information ek is used to encoder latent dynamics zk. Next, the historical data is employed to reconstruct the latent dynamics. Lastly, the further motion state is predicted via state net module. Several tests are arranged to preliminarily validate the effectiveness.
Questions:
(1)	Although data-based methods can be very effective, the quality of their results depends on the quality of the data used for identification. In this case, the data used came from a system like quadrotor with a specific closed-loop control parameters, and is therefore only effective under these conditions. If the gains change (for example, during adjustments from the ground station), does the offline training model need to be re-obtained? The generalization in unseen environments should be provided in the manuscript.
[1] Feedback favors the generalization of neural ODEs. ICLR
[2] Millimeter-level pick and peg-in-hole task achieved by aerial manipulator. IEEE TRO
(2)	Experiments. 1) In 2D wheel robot: overly simplifies the friction settings for the movement of the vehicle in different textured environments. 2) In 3D quadrotor: real wind disturbance is three-dimensional and time-varying. It is not sufficient to consider only the constant x-direction in simulation. It is better to provide details in real fight tests. 3) The purpose of the algorithm is to improve the dynamic tracking performance by accurately predicting the future motion state. It is not sufficient to merely present the results in a table. Providing more experiment details, such as pictures and more aggressive references.
(3)	Comparison: RL-based approaches are frequently mentioned, but it has not been compared in the tests. Moreover, it is also necessary to compare some classic approximate methods including adaptive NN, disturbances observer.
[3] Precise end-effector control for an aerial manipulator under composite disturbances: Theory and experiments. IEEE TASE
(4)	In section 4.1, the authors indicate that it is effective to align two domains together in latent space even historical data carries distinct physical meanings and have significantly different dimensionality compared to ek.  Although a similar approach is proven in image field, I have some doubts about this during the process of learning dynamics. Provide more details.
[4] Image style transfer using convolutional neural networks. CVPR
(5) Please quantify the impact of the two parts on the accuracy of the final state prediction. Please quantify the impact of the two parts on the accuracy of the final state prediction.

**Strengths:**

An ode-based two-phase training procedure is used to learn latent environment representations for improving system dynamic tracking performance. In Phase 1, privileged environmental information ek is used to encoder latent dynamics zk. Next, the historical data is employed to reconstruct the latent dynamics. Lastly, the further motion state is predicted via state net module. Several tests are arranged to preliminarily validate the effectiveness

**Weaknesses:**

Theoretically, the issue of generalization is not considered and the experimental setup is insufficient.

**Questions:**

Questions: (1) Although data-based methods can be very effective, the quality of their results depends on the quality of the data used for identification. In this case, the data used came from a system like quadrotor with a specific closed-loop control parameters, and is therefore only effective under these conditions. If the gains change (for example, during adjustments from the ground station), does the offline training model need to be re-obtained? The generalization in unseen environments should be provided in the manuscript. [1] Feedback favors the generalization of neural ODEs. ICLR [2] Millimeter-level pick and peg-in-hole task achieved by aerial manipulator. IEEE TRO (2) Experiments. 1) In 2D wheel robot: overly simplifies the friction settings for the movement of the vehicle in different textured environments. 2) In 3D quadrotor: real wind disturbance is three-dimensional and time-varying. It is not sufficient to consider only the constant x-direction in simulation. It is better to provide details in real fight tests. 3) The purpose of the algorithm is to improve the dynamic tracking performance by accurately predicting the future motion state. It is not sufficient to merely present the results in a table. Providing more experiment details, such as pictures and more aggressive references. (3) Comparison: RL-based approaches are frequently mentioned, but it has not been compared in the tests. Moreover, it is also necessary to compare some classic approximate methods including adaptive NN, disturbances observer. [3] Precise end-effector control for an aerial manipulator under composite disturbances: Theory and experiments. IEEE TASE (4) In section 4.1, the authors indicate that it is effective to align two domains together in latent space even historical data carries distinct physical meanings and have significantly different dimensionality compared to ek. Although a similar approach is proven in image field, I have some doubts about this during the process of learning dynamics. Provide more details. [4] Image style transfer using convolutional neural networks. CVPR (5) An ode-based two-phase training procedure is used to learn latent environment representations for improving system dynamic tracking performance. In Phase 1, privileged environmental information ek is used to encoder latent dynamics zk. Next, the historical data is employed to reconstruct the latent dynamics. Lastly, the further motion state is predicted via state net module. Several tests are arranged to preliminarily validate the effectiveness. Questions: (1) Although data-based methods can be very effective, the quality of their results depends on the quality of the data used for identification. In this case, the data used came from a system like quadrotor with a specific closed-loop control parameters, and is therefore only effective under these conditions. If the gains change (for example, during adjustments from the ground station), does the offline training model need to be re-obtained? The generalization in unseen environments should be provided in the manuscript. [1] Feedback favors the generalization of neural ODEs. ICLR [2] Millimeter-level pick and peg-in-hole task achieved by aerial manipulator. IEEE TRO (2) Experiments. 1) In 2D wheel robot: overly simplifies the friction settings for the movement of the vehicle in different textured environments. 2) In 3D quadrotor: real wind disturbance is three-dimensional and time-varying. It is not sufficient to consider only the constant x-direction in simulation. It is better to provide details in real fight tests. 3) The purpose of the algorithm is to improve the dynamic tracking performance by accurately predicting the future motion state. It is not sufficient to merely present the results in a table. Providing more experiment details, such as pictures and more aggressive references. (3) Comparison: RL-based approaches are frequently mentioned, but it has not been compared in the tests. Moreover, it is also necessary to compare some classic approximate methods including adaptive NN, disturbances observer. [3] Precise end-effector control for an aerial manipulator under composite disturbances: Theory and experiments. IEEE TASE (4) In section 4.1, the authors indicate that it is effective to align two domains together in latent space even historical data carries distinct physical meanings and have significantly different dimensionality compared to ek. Although a similar approach is proven in image field, I have some doubts about this during the process of learning dynamics. Provide more details. [4] Image style transfer using convolutional neural networks. CVPR (5) Please quantify the impact of the two parts on the accuracy of the final state prediction. Please quantify the impact of the two parts on the accuracy of the final state prediction.

---

> ### Author Response · Authors · 2025-12-03
>
> **Generalize across control gains and environments**
>
> We appreciate the reviewer’s concern. We agree that if the low-level controller gains change, the induced closed-loop behavior of the system also changes. This issue is not unique to our method and is commonly observed in most data-driven dynamics learning approaches. In practice, if changes in the low-level PID parameters are also of interest, this limitation can be mitigated by treating the PID parameters as part of the environmental factors to be inferred or conditioned on during training.
>
> In addition, the results of the unseen environments are provided in the quadrotor simulator (Table 3 in the manuscript). During data collection, we sample the disturbance force in the range of [−1, 1] N and collect trajectories in each wind field. During testing, the quadrotor is subjected to wind fields beyond the [−1, 1] N range, representing beyond range environmental conditions.
>
> **Concerns about simplified simulation environments**
>
> **2D wheeled robot:**
> We thank the reviewer for the suggestion. The friction model is intentionally simplified because the goal of this experiment is to evaluate the controller’s performance under representative surface variations, rather than to fully reproduce granular terrain physics. In the simulation, instead of using a pure unicycle model, we explicitly model wheel–ground contacts and include cases where the two wheels experience different surface textures. We believe this level of detail is sufficient for assessing our method, and the conclusions would likely remain unchanged even with more complex friction models.
>
> **3D quadrotor:**
> We thank the reviewer for the comment. While real wind is three-dimensional and time-varying, our objective is to evaluate disturbance sensitivity rather than to replicate full atmospheric dynamics.
> We want to clarify that we include both spatially and temporally varying wind fields, so the disturbance is not constant. One of our test cases includes spatial variation along the x-direction combined with a time-varying 3D wind disturbance, demonstrating the controller’s ability to handle both spatially and temporally varying wind conditions, as one might encounter in a real flight test when a fan is blowing on a quadrotor. We have also added a new test case with a quadrotor under 3D disturbance in the general response, and the results show that our algorithm can also perform well in this setting. We acknowledge that including real-flight tests could further strengthen the results. However, given limited resources and our goal of comparing algorithms in a controlled setting, we believe that the environmental variations in our simulated test cases are sufficient for assessing our method. Extending the evaluation to real-flight tests would be a good direction for future work.
>
> **More experiment details**
>
> We thank the reviewer for the suggestion. In the updated PDF, we have included additional real-world experimental pictures in Section 6 and the trajectories of our model and baselines for the path-tracking tasks in Section A.8. We have also added a more aggressive reference track in the real-world experiment. Please see the results for this additional reference in the general response.
>
> **Comparison with traditional adaptive control**
>
> We thank the reviewer for the comment. To ensure a fair comparison, we primarily evaluate our approach against other learned dynamics methods. Comparing our method with classic equation-based approaches is challenging because such methods typically assume a well-known system dynamics model, whereas our approach does not rely on this assumption.
>
> **More details on latent-space alignment**
>
> We thank the reviewer for the question. Here are more details. Aligning the two domains in latent space allows us to design two different network structures for processing the distinct types of data. Specifically, we use an MLP as the backbone for the encoder in Phase 1 and a CNN as the backbone for the adaptive module in Phase 2. This design decomposes the complex mapping problem into two parts: the MLP focuses on mapping to the latent space, while the CNN handles time-series analysis. Processing different data structures with different model architectures and aligning them in the same latent space has been found to be effective in multi-modality settings in both vision and robotics. In our case, it helps learning even when historical data carry distinct physical meanings and have significantly different dimensionality compared to ek.
>
> **Quantify the impact of the two parts on the accuracy of the final state prediction**
>
> We thank the reviewer for the comment. Due to page limitations, the impact of Phase 1 and Phase 2 on the accuracy of state prediction is presented in Section A.4.3, where we compare the dynamics learned in both phases along with other baselines.

---

### Official Review · Reviewer_qTNt · 2025-10-30

**Soundness:** 3
**Presentation:** 3
**Contribution:** 2
**Rating:** 6
**Confidence:** 4

**Summary:**

The paper proposes adaptive dynamics NODE (AD-NODE), a learned dynamics model for model predictive control (MPC) of mobile robots that can adapt to environmental changes at runtime. The dynamics are modelled as neural ordinary differential equations (NODE) that can be numerically integrated (e.g. forward Euler) to obtain the system state. The authors take inspiration from rapid motor adaptation (RMA) and train a latent embedding representing the current environment condition that is given to the NODE network as input. To better handle out-of-distribution parts of the state space, online fine-tuning of the learned dynamics model based on recorded trajectories from an experience replay buffer is included. The performance of AD-NODE is validated on a simulated and real differential drive robot and on a simulated quadrotor drone. AD-NODE is compared to several baselines and improvements in control performance are demonstrated.

**Strengths:**

* AD-NODE combines two known approaches from the literature, NODE as a dynamics model and RMA for adaptation to changing environment conditions, in an incremental but novel way and uses it for a sampling-based MPC.
* The paper validates AD-NODE for two robotic systems in simulation and real world experiments and showcase an improvement in performance, especially compared to NODE without adaptation.
* Their proposed online fine-tuning of the dynamics helps with out-of-distribution cases at test time in the simulated quadrotor control experiments.
* The paper reports an MPC planning time of less than 0.01 seconds for several hundred parallel planning roll-outs, highlighting the real-time capability of their approach.

**Weaknesses:**

* In the related work section, the paper misses works on dynamics model learning for navigation from the robotics community such as
  - Guttikonda et al. Context-Conditional Navigation with a Learning-Based Terrain- and Robot-Aware Dynamics Model. ECMR 2023.
  - Vertens et al. Improving deep dynamics models for autonomous vehicles with multimodal latent mapping of surfaces. IROS 2023.
* In figure 2.b): Why does the state-net output x_k+1 in phase 1 and x_k in phase 2? Shouldn't this be x_k in both cases?
* Experiments are conducted with simplified robot models in simulation or a differential drive robot in a simplified setup on a flat table with artificial friction changes. The paper should discuss limitations and assumptions and how they could be addressed to scale to more complex real robots like wheeled robots with suspension or more complex terrain interactions including soft or granular terrain. The model-free RL approach RMA demonstrates applicability in more realistic settings, e.g., it evaluates RMA extensively in various outdoor scenarios for a quadruped robot.
* When online finetuning the dynamics model, why does the parametrization by the latent vector not diverge from the learned mapping of state history to environment properties?
* Sec. 5.4, why is the wind limited to the x-direction? Does it work with wind in 2D ?
* Presenting results with 2 or 3 digits after the floating point should be sufficient in most cases.
* How is the friction coefficient measured in the real robot experiment in Sec. 6.2 ?
* Sec 6.3, please explain the main ideas of the cost function design in the main paper.

**Questions:**

* Please address the points raised in the "weaknesses" section.

---

> ### Author Response · Authors · 2025-12-03
>
> **Expand the related works:**
>
> We thank the reviewer for the suggestion. We have added the relevant works in the revised Related Work section to make the survey more comprehensive.
>
> **Typo in Figure 2.b:**
>
> We thank the reviewer for catching this. We have corrected Figure 2b and the corresponding text in the revised manuscript.
>
> **Discussion on scaling to more complex real robots or more complex terrain interactions:**
>
> We thank the reviewer for the comment. To scale to more complex environments, we can expand the dimensionality of the environmental factors to incorporate parameters from richer contact models that capture heterogeneous friction, soft or granular terrain, and other effects. Once these factors are defined, our existing framework and training pipeline can be used to inform the model about more complex terrains.
>
> For more complex robots, unactuated components (e.g., suspension) can be learned implicitly by the dynamics model, while actuated components with additional degrees of freedom can be accommodated by expanding the state representation. These considerations demonstrate that our approach can generalize to more realistic robots and terrains while maintaining the core advantages of our framework.
>
> **Divergence of the latent vectors during online learning:**
>
> We thank the reviewer for the question. During online finetuning, the robot is deployed on a single surface whose environmental properties are known beforehand. This allows us to finetune the adaptive module by regressing the environmental latent vector toward its ground-truth value, preventing the model from drifting away from the learned mapping between state history and environment factors. After this finetuning stage, we deploy the robot in more complex scenarios involving multiple surfaces.
>
> **Concerns regarding higher-dimensional wind fields:**
> We thank the reviewer for the question. Please see the general response for the discussion of the 3D wind-field disturbance. We have also included the latest results in the updated PDF.
>
> **Presenting results with 2 or 3 digits after the decimal point should be sufficient in most cases:**
> We thank the reviewer for the suggestion. We have revised the reported numbers accordingly in the updated PDF.
>
> **Friction coefficient measurement:**
> We thank the reviewer for the question. In our real-robot experiments, we approximate friction coefficients by pushing and rotating a plastic cube on each surface and recording its motion. This method provides relative friction values across the different surfaces.
>
> **Explain the main ideas of the cost function design in the main paper:**
>
> We thank the reviewer for the question. The main ideas behind the cost function designs are as follows:
> • **Differential-drive robot and real-world deployment:** We design two task-specific cost functions. For goal reaching, J1 penalizes deviations in both velocity and heading, where θkpp denotes the pure-pursuit heading toward the goal. For path tracking, J2 penalizes velocity, position, and reference-heading errors.
> • **Quadrotor 3D pose tracking:** For general 3D tracking, J3 penalizes both position and orientation errors, with the quaternion term providing a smooth and sign-invariant measure of orientation deviation.
> Overall, these cost functions encode the desired behaviors through weighted penalties on predicted future states.

---

### Official Review · Reviewer_fe1Y · 2025-11-01

**Soundness:** 4
**Presentation:** 4
**Contribution:** 1
**Rating:** 0
**Confidence:** 5

**Summary:**

The paper proposes AD-NODE, an adaptive dynamics learning framework for mobile robot control that integrates Neural ODE with Model Predictive Control (MPC). The method infers hidden environmental variations (e.g., terrain friction, wind) from state–action history rather than explicit sensors, enabling continuous-time adaptation. AD-NODE is trained in two phases: learning dynamics with privileged environmental information, and reconstructing latent environmental embeddings from history for partially observed settings. The approach is validated on a 2D differential wheeled robot, a 3D quadrotor, and a real Sphero BOLT robot, showing superior performance in both goal-reaching and path-tracking tasks under spatially and temporally varying conditions compared to CaDM, meta-learning, and fixed NODE baselines

**Strengths:**

- Clear Motivation for Adaptive Dynamics. The paper articulates a strong motivation: existing model-based controllers struggle under unmodeled environmental variations, and most model-free approaches (e.g., RMA) are data-hungry. Section 1 convincingly situates AD-NODE as a bridge between adaptive RL and continuous-time modeling

**Weaknesses:**

- While the paper addresses adaptive dynamics modeling for model-based control, similar ideas have recently appeared in AnyCar to Anywhere: Learning Universal Dynamics Model for Agile and Adaptive Mobility (ICRA 2025), which also employs a Transformer-based dynamics model that takes state–action history to predict future states and achieve agile mobility in the wild. Compared to that line of work, the present submission offers limited methodological novelty in both the dynamics learning and controller design; the main contribution appears to be an integration of existing modules (NODE + MPC) into a coherent system. As such, the paper’s emphasis may align more closely with robotics systems or planning venues rather than introducing a fundamentally new learning or control algorithm.

**Questions:**

- Given the focus on mobile robots, can the authors comment on whether the NODE-based dynamics would remain tractable for higher-dimensional systems like quadrupeds or manipulators, where the ODE solver overhead grows rapidly?

---

> ### Author Response · Authors · 2025-12-03
>
> **Concern about similarity to existing transformer-based dynamics model:**
>
> We thank the reviewer for pointing out this paper. For clarification on the connection and differences between the two works, please refer to the general response.
>
> **NODE on higher-dimensional system:**
>
> We thank the reviewer for the thoughtful question. For higher-dimensional systems such as quadrupeds or manipulators, the tractability of NODE-based dynamics primarily depends on the cost of the MLP forward pass rather than the ODE solver itself.
> During inference, our model performs a single MLP evaluation to compute the state derivative, followed by a forward Euler step, so the computational time will mainly depend on how fast the forward pass of MLP is.
>
> During training, we acknowledge that NODE may experience exploding or vanishing gradients in higher-dimensional settings. As noted in Section 4.3, we mitigate this issue through curriculum learning, which stabilizes training by gradually increasing trajectory length. Empirically, this strategy allows NODE training to remain stable and tractable even for higher-dimensional systems.

---

### Author Response · Authors · 2025-12-03
**General response**

We thank the reviewers for highlighting AD-NODE’s clear motivation as a bridge between adaptive RL and continuous-time modeling. Reviewers found the experiments comprehensive, noted the effectiveness of online fine-tuning for out-of-distribution quadrotor scenarios, and recognized fast MPC planning as evidence of real-time feasibility. Below, we summarize the main reviewer concerns and our responses:

**Concern about similarity to existing transformer-based dynamics models:**

We thank the reviewer for pointing out the paper *AnyCar to Anywhere*. While our work might appear similar to the paper, as both approaches aim to learn dynamics for model-based controllers, our method takes a fundamentally different approach. The cited work uses Transformers as dynamics models and infers environmental information implicitly by predicting the next state directly from state–action histories in an end-to-end manner. However, learning such mappings is challenging due to the high dimensionality of the histories and the difficulty of ensuring that the model captures the true environmental factors.

While they apply Transformer to address the challenge, our method learns the dynamics in two phases and leverages simulation data with privileged information to guide the learning of environmental conditions. This leads to a more interpretable framework and lightweight adaptation mechanism, suitable for onboard computing. Moreover, using a Neural ODE backbone introduces an inductive bias that improves long-term accuracy without increasing the size of dataset or model. The design choice is different from sequence-based Transformer formulation, which may require a large dataset and more computational resource to train it. Finally, while the cited work validates on multiple car platforms, we demonstrate AD-NODE on a differential-wheeled robot, a quadrotor simulator, and a real-world spherical robot, highlighting that our work focuses more on proposing a general framework that can be applied to different robotic domains. We have added this work to the related work section to acknowledge these connections and distinctions.

**Comparison with model-free RL methods including RMA:**

We add two representative model-free baselines for the quadrotor simulator: RMA and DATT. Based on the results, we believe that model-based and model-free methods offer complementary strengths. Model-based approaches, which learn a dynamics model from offline or online data and use planners like MPPI to incorporate desired trajectories and constraints (e.g. collision avoidance) explicitly in their cost functions, enabling safer behavior and reducing the need for trajectory-specific training data. However, they incur higher computational cost at inference. Model-free methods such as PPO train policies directly from on-policy data, making them simpler to deploy and much faster at runtime, but their performance can degrade when training trajectories do not fully cover evaluation conditions, and they cannot easily incorporate constraints. Overall, model-based methods provide flexibility and safety, while model-free methods offer speed and efficiency. We have also updated our PDF to include the results and the discussion in the experimental section, and the qualitative results are also included in Section A.8.1.

**A more aggressive reference in real-world experiments:**

To better demonstrate our model’s capabilities, we test the Sphero BOLT robot on a small triangular reference track to showcase our model's ability to perform aggressive turns. The setup and layouts are shown in Figure 5 of the updated PDF. The results indicate that the improvements persist in the triangle-tracking environment across two friction layouts, yielding smaller tracking errors. We have also updated the PDF to include these results and their discussion, with additional qualitative results presented in Section A.8.2.

**Additional qualitative results, experimental details, and videos:**

We have updated the PDF to include additional qualitative results, such as trajectories of our model and baselines for path tracking across different environments (see experimental section and Section A.8), as well as more images of the real-world experimental setup in Section 6. The videos are available at: https://www.youtube.com/playlist?list=PLOBu3KsQCwsQYfdrl4DlCVNuh_eNzGTwW

**More complex 3D wind field:**

We have added new results in Section A.7, where we generate a 3D wind field using a Brownian motion model and test all algorithms under this more complex setting. Our method continues to outperform the baselines, demonstrating its ability to scale to challenging environments and maintain the performance trends reported in Table 3. While the wind field in Table 3 is directional, layout 2 reflects a more realistic scenario in which a fan produces a wind field that is roughly aligned, modeling with spatial variation in the x-direction and 3D disturbances following the formulation used in DATT.

---

### Meta-Review · Area_Chair_9PZb · 2026-01-07

**Summary:**

The primary concerns driving the rejection recommendation center on novelty, experimental realism, and baseline comparisons.
1. Limited Novelty (Reviewer fe1Y): Reviewer fe1Y strongly criticized the work for lacking methodological novelty, characterizing it as a straightforward integration of existing modules (Neural ODEs + MPC + RMA-style adaptation) rather than a fundamental contribution.
2. Simplified Experimental Settings (Reviewer qTNt, zJvQ): Multiple reviewers noted that the experiments were oversimplified. Reviewer qTNt and zJvQ pointed out that the "real-world" experiments used artificial friction on flat tables and the simulation used constant or 1D wind, which fails to capture the complexity of real robotic deployment (e.g., 3D turbulence, granular terrain, suspension dynamics).
3. Insufficient Baselines & Generalization (Reviewer zJvQ, igpy): Reviewers noted the absence of critical comparisons to Model-Free RL (specifically RMA) and classic adaptive control in the initial submission. Reviewer zJvQ also raised valid concerns about the model's inability to generalize if low-level control gains change.

**Reviewer Concerns:**

- Concerns Addressed by Rebuttal
  - Baselines: The authors added comparisons to RMA and other RL baselines in the general response, addressing the specific request from Reviewers zJvQ and igpy.
  - Technical Clarifications: The authors clarified the scalability of NODE to high-dimensional systems (answering Reviewer fe1Y) and explained the latent alignment strategy (answering Reviewer zJvQ).
  - Experimental Details: Issues regarding specific cost functions, and friction measurement methods were resolved.
- Outstanding Concerns
  - Fundamental Novelty: The rebuttal did not effectively counter Reviewer fe1Y's core argument that the paper is a system integration paper rather than a learning/control algorithm breakthrough.
  - Experimental Realism: While the authors argued that their simplified friction and wind models are "sufficient," this defense is weak against the reviewers' desire for rigorous validation in complex, unstructured environments. The reliance on simplified proxies (e.g., plastic cubes for friction, 1D wind components) undermines the claim of robust adaptive control.
  - Generalization Limits: The authors admitted that changes in low-level control gains would require retraining or adding gains to the state space, confirming Reviewer zJvQ's concern about the fragility of the learned dynamics regarding system parameter changes.

**Reviewer Scores:**

Reviewer fe1Y would likely maintain the score of 0 or raise the score to 1, as the rebuttal failed to refute the primary critique regarding the lack of methodological novelty and the similarity to concurrent work.

Reviewer qTNt would likely maintain the score of 6, as Reviewer qTNt were generally positive about system integration.

Reviewer zJvQ would likely maintain their score of 4, as the authors' admission regarding the sensitivity to control gains and the defense of simplified simulation environments confirms the reviewer's suspicions about limited generalization.

Reviewer igpy might raise their score to 6 due to the inclusion of the RMA baseline.

---

### Decision · Program_Chairs · 2026-01-26

Reject